# SUBER: An RL Environment with Simulated Human Behavior for Recommender Systems

## Abstract

Reinforcement learning (RL) has gained popularity in the realm of recommender systems due to its ability to optimize long-term rewards and guide users in discovering relevant content. However, the successful implementation of RL in recommender systems is challenging because of several factors, including the limited availability of online data for training on-policy methods. This scarcity requires expensive human interaction for online model training. Furthermore, the development of effective evaluation frameworks that accurately reflect the quality of models remains a fundamental challenge in recommender systems. To address these challenges, we propose a comprehensive framework for synthetic environments that simulate human behavior by harnessing the capabilities of large language models (LLMs). We complement our framework with in-depth ablation studies and demonstrate its effectiveness with experiments on movie and book recommendations. By utilizing LLMs as synthetic users, this work introduces a modular and novel framework for training RL-based recommender systems. The software, including the RL environment, is publicly available.[1]

## 1 Introduction

In an age defined by the ubiquitous presence of digital platforms in both leisure and commerce, recommender systems have emerged as instrumental tools in guiding user choices. From Netflix tailoring movie suggestions to match cinematic tastes of users to Amazon presenting personalized lists of products to shoppers, recommendation systems are the engines driving enhanced user experiences and platform engagement (Steck et al., 2021; Agrawal et al., 2023).

Reinforcement Learning (RL), with its principles rooted in learning by interaction, provides a compelling approach to dynamically and adaptively tailor recommendations. Recommender systems should take into account both short and long term rewards and direct the interests of users towards appropriate recommendations. An increasing body of research has investigated the use of RL in recommender systems (Ie et al., 2019b; Chen et al., 2019a; Liu et al., 2022; Afsar et al., 2022; Lin et al., 2023). Although promising, the use of RL for recommendation systems comes with its own set of challenges:

*Data Availability*: RL algorithms require a significant amount of data from interactions with the environment to learn effective policies. However, in the case of recommender systems, users may quickly abandon the service if they receive random or irrelevant recommendations. This makes it impractical to collect the large amount of data needed to train an RL model without compromising the user experience (Zhang et al., 2016).

*Unknown user model*: In RL, a reward function is crucial to allow the model to learn effectively. In the context of recommender systems, designing an appropriate synthetic reward function that accurately reflects user satisfaction or preferences can be challenging due to the complexity of modeling human behavior (Chen et al., 2019b; Shi et al., 2019).

*Model evaluation*: A key challenge in recommender systems is the evaluation of models without directly interacting with real users, thus avoiding any potential negative impact on the user experience.

---

[1] https://github.com/anonymous-suber/code

On the other hand, evaluating on offline data does not guarantee good recommendation performance in the real world (Shani & Gunawardana, 2011; Garcin et al., 2014).

In this work, we propose a "Simulated User Behavior for Recommender Systems" (SUBER), a novel framework for recommender systems to address the aforementioned challenges. SUBER is a framework for synthetic environments that use Large Language Models (LLM) at their core. SUBER leverages recent advances in LLMs to simulate human behavior Park et al. (2023); Argyle et al. (2023). Furthermore, by training on large amounts of data, LLMs have obtained inherent knowledge about movies, books, and various other objects. These strengths, the ability to mimick human behavior coupled with vast knowledge about humanity, uniquely position LLMs as a powerful tool to simulate users in synthetic environments for recommender systems. Therefore, SUBER serves as a versatile playground for researchers, allowing them to experiment with different LLM configurations, fine-tune user specifications, and improve their RL strategies. Our contributions can be summarized as follows:

- We propose SUBER, a general framework to train and evaluate RL-based recommender systems. Our framework consists of a gym environment containing an LLM that is capable of simulating human behavior and rate recommended items accordingly.
- We conduct extensive ablation studies to examine the influence of the different components in our proposed framework. Moreover, we present findings across several families of LLMs and their effect on the performance of the environment, highlighting the effectiveness of various LLMs in replicating human choices for item recommendations.
- We experimentally assess our environment by evaluating both a movie recommendation setting and a book recommendation setting. We also make all code available as open-source.

## 2  RELATED WORK

**Recommender systems.** Platforms such as YouTube (Ie et al., 2019b; Chen et al., 2019a) and BytePlus (Liu et al., 2022) are two of many recent successful examples of training and evaluating recommender systems with online data. Traditional and neural recommender systems have been researched extensively over the past three decades (Goldberg et al., 1992; Su & Khoshgoftaar, 2009; Bobadilla et al., 2013; Shi et al., 2014; Lu et al., 2015; Zhang et al., 2019). However, since our work focuses on RL in recommender systems (RL4Rec), we limit the related work to this area of research. While RL4Rec has been the subject of several studies, a majority of the work has relied predominantly on training and evaluating based on offline datasets (Afsar et al., 2022; Lin et al., 2023). As indicated by Afsar et al., online assessment is the preferred approach for evaluation. However, it presents significant challenges with respect to complexity and expense. In contrast, offline evaluation takes place in a static and biased environment. Afsar et al., therefore, call for creating a versatile simulator for RL4Rec similar in nature to OpenAI's Gym for conventional RL tasks (Brockman et al., 2016). Additional challenges exist in the wider domain of RL, specifically regarding issues related to off-policy learning and offline policy evaluation, which become even more complex when incorporated into recommender systems (Precup et al., 2001; Gelada & Bellemare, 2019; Kumar et al., 2019).

Notable efforts have been made to address the limitations of offline learning in recommender systems. To this end, many simulation environments for recommender systems have been developed. Rohde et al. (2018) presented RecoGym, a synthetic environment that addresses exploding variance by simulating user responses to different recommendation strategies. RecSim (Ie et al., 2019a) is a customizable synthetic simulation platform that incorporates various assumptions about user preferences, item familiarity, user latent states and dynamics, and choice models. Chen et al. (2019b) proposed a generator that captures the underlying distribution of historical interactions of users and learns to generate realistic interactions. Extending this idea, Shi et al. (2019) proposed Virtual-Taobao, a virtual shopping environment and demonstrated the superiority of policies developed in this framework over traditional supervised techniques in real-world settings. Wang et al. (2023a) introduced the RL4RS dataset to address the lack of validated simulation environments and advanced evaluation methods in RL-based recommender system research. The dataset is collected from a NetEase game and anonymized through a three-step process. Zhao et al. (2023) propose KuaiSim, a versatile environment that provides user feedback with multi-behavior and cross-session

Table 1: Comparison of simulation environments for recommender systems. We list whether the user and item dataset are real or synthetic. Simulation Engine indicates the different approaches used. For the evaluation strategy, we distinguish between offline evaluation on the original dataset used to train the simulator, online testing on a platform, sanity checks, and case studies.

| Paper | User dataset | Item dataset | Simulation engine | Evaluation strategy |
|---|---|---|---|---|
| Adversarial (Chen et al., 2019b) | Real | Real | GAN | Offline |
| VirtualTaobao (Shi et al., 2019) | Real | Real | GAN | Online |
| RL4RS (Wang et al., 2023a) | Real | Real | Transformer | Online |
| KuaiSim (Zhao et al., 2023) | Real | Real | Transformer | Offline |
| RecoGym (Rohde et al., 2018) | Synthetic | Synthetic | Statistical modelling | Sanity checks |
| RecoSim (Ie et al., 2019a) | Synthetic | Synthetic | Statistical modelling | Case studies |
| SUBER (our) | Synthetic | Real | LLM | Sanity checks and case studies |

responses, supporting three tasks: request-level list-wise recommendation task, whole-session-level sequential recommendation task, and cross-session-level retention optimization task. Unlike previous approaches, our work leverages natural language by using LLMs to simulate user behavior. In addition, our framework is not dataset dependent, and therefore, the set of users and items are not restricted to specific domains.

**Large Language Models.** There have been significant recent advances in the field of LLMs. These models are primarily based on the transformer architectures introduced by Vaswani et al. (2017) and have continued to grow in size, capability, and performance. The Generative Pre-trained Transformer (GPT) series by OpenAI (Brown et al., 2020; OpenAI, 2023) is one of the most notable developments in this area, demonstrating the immense potential and scalability of transformer-based models. The recent release of foundation language models such as Llama-1 and Llama-2 (Touvron et al., 2023a;b), has democratized the access to these large LLMs. This has paved the way for the creation of instruction-following models such as Vicuna (Zheng et al., 2023) and Alpaca (Taori et al., 2023). Meanwhile, numerous efforts have focused on optimizing the memory consumption and inference speed of LLMs. For example, GPTQ (Frantar et al., 2022) compressed model parameters to 4 bits, allowing larger models to run on hardware with less memory and without significant loss of performance.

LLMs can generate textual content that rivals the quality of human-generated text (Brown et al., 2020). However, their applications go beyond text generation. Park et al. (2023) demonstrated how LLMs can be used to simulate human behavior. These simulated agents wake up, cook, go to work, make decisions, and reflect on past experiences in a believable manner. Furthermore, Argyle et al. (2023) suggests using language models as surrogates for certain demographic groups within social science research. Their study demonstrates how conditioning GPT-3 on the socio-demographic backgrounds of real human subjects can accurately replicate response distributions among diverse human subgroups.

Contemporary work has also integrated LLMs into recommender systems. Kang et al. (2023) demonstrated that fine-tuned LLMs outperform traditional supervised methods in predicting user ratings with less training data, while Wang et al. (2023b) employed LLMs as a recommendation agent, showcasing their potential in enhancing recommender systems. Both works show how LLMs can act as a good predictor of the ratings that a user would assign to an item. The authors further investigated whether LLMs can also be used as a recommender directly; they restricted their experiment to choosing an item from a list of 100 items. However, this task is still challenging for LLMs, as they must have knowledge of the entire set of possible items to be recommended. The limited context length does not allow one to provide a list of all possible items in the prompt to an LLM. Therefore, until now, LLMs have not been able to overcome traditional recommender systems. Compared to these contemporary works, our main difference is the integration of LLMs as simulation environments for item recommendation, while related works train the LLM to be the recommender system itself.

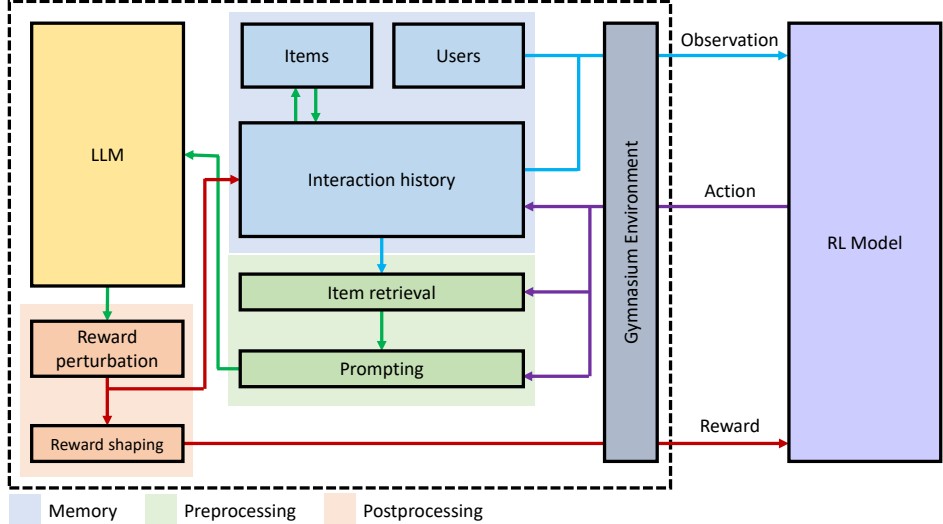

Figure 1: Overview of SUBER. The environment is built as a modular framework where each component can be modified as required. The basic control flow is as follows: The environment provides an observation using the memory module; the RL model returns an item recommendation in the form of an action, which is processed into a prompt by the memory and preprocessing component before being passed to the LLM. The score returned by the LLM is postprocessed, stored in memory and returned as a reward to the RL model.

## 3 FRAMEWORK

To address the aforementioned challenges of data availability, unknown user model, and model evaluation, we propose SUBER, an environment designed to simulate human behavior through the integration of LLMs. SUBER serves a dual purpose by generating synthetic data and harnessing the capabilities of LLMs to replicate the behavior of individuals with unknown patterns. Additionally, this dynamic environment can function as a model evaluation tool for recommender systems.

SUBER consists of an LLM component and three separate modules that contain multiple individual components. An overview of the overall structure is presented in Figure 1. The internal memory module of the environment contains two separate datasets, one for users and one for items. The environment also includes a preprocessing module that retrieves raw data from the memory module and transforms it to ensure compatibility with the LLM. Finally, a postprocessing component transforms the output produced by the LLM before returning it to the RL model.

The interaction with an RL model involves the following information flow: initially, the environment selects a user from memory, along with their interaction history (i.e., items and associated ratings) as the observation for the RL model. The RL model then recommends an item to the user as its action, with an action space equal to the number of items in the environment. The action and observation are subsequently processed through the preprocessing module, the LLM component, and the postprocessing module. Finally, the environment returns a reward corresponding to the postprocessed rating predicted by the LLM. We describe each module in more detail in the following sections.

Our environment is designed with easy accessibility and extensibility in mind. Therefore, we chose a modular approach and based the environment interface on the standardized API of Gymnasium (Towers et al., 2023). Different components can be modified at will, providing additional flexibility in future design choices.

### 3.1 MEMORY

We introduce the following notation. We define $U$ as the set of users and $I$ as the set of items. For every user-item pair $(u, i) \in U \times I$, we have a set $R_{u,i}$ that records all interactions between user $u$

and item $i$. Similarly, for every user $u$ we define with $R_u$ the set of all interactions with all movies, defined as follows:

$$R_u = \{(i, h)|i \in I, h \in R_{u,i}\}. \tag{1}$$

The memory module consists of three components: an item dataset, a user dataset, and a record of all interactions between users and items. This interaction history stores the set of interactions $R_{u,i}$ for each user-item pair $(u, i)$. Every interaction between the RL model and the environment produces a new interaction record between a user and an item, which is appended to the interaction history.

## 3.2 PREPROCESSING

**Item retrieval.** As the RL model interacts with the environment, the interaction history grows. It may be challenging to extract relevant information from long histories, and increasing history length will likely exceed the context length of current LLMs (Park et al., 2023). To address this issue, we propose an item-retrieval component responsible for retrieving the most appropriate items for the current query from the interaction history of a user. Additionally, as user interests and preferences may evolve over time, relying solely on user features may not accurately capture current interests. Therefore, historical rating data are used to provide a more detailed depiction of their evolving preferences.

**Prompting.** The prompting component aggregates the information retrieved by the item retrieval component, creating a prompt that contains the necessary details for the LLM, including the user and query item data. The objective of this prompt is to enable the LLM to accurately predict the rating of the current query item. An example of such a prompt as part of an interaction example can be seen in Figure 2.

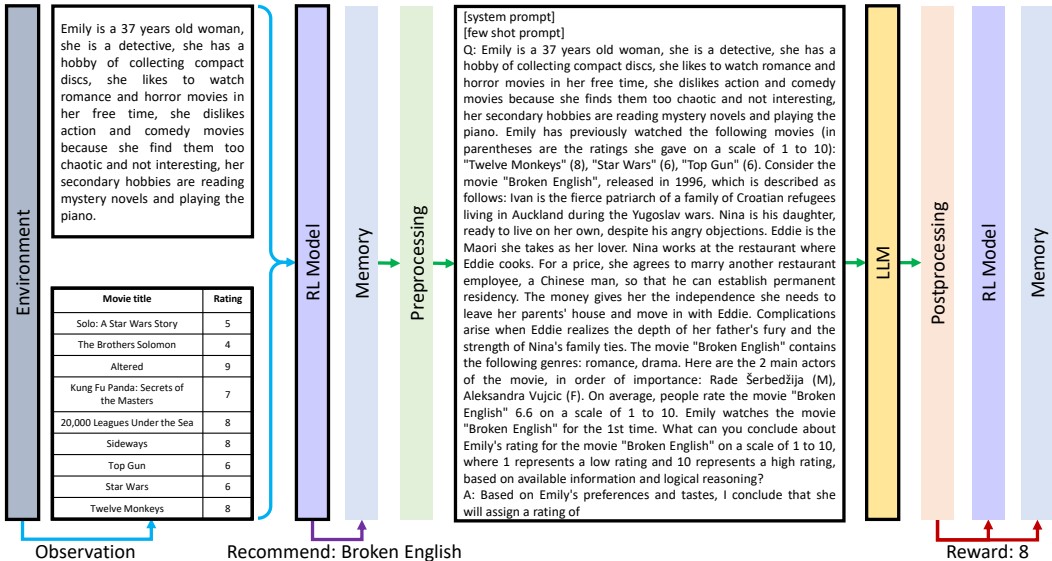

Figure 2: Pipeline of one interaction between the RL model and SUBER. The environment provides an observation in the form of a user description and user-item interaction history to the RL model. The RL model then recommends an item, which is processed into a prompt together with the user description and interaction history. The LLM uses this prompt to generate a reward for the recommended item. The reward is stored as part of the user-item interaction history and returned to the RL model.

## 3.3 POSTPROCESSING

**Reward perturbation.** The reward perturbation component introduces noise into the ratings generated by the LLM. This component functions as a simulation of "concept drift" for users (Žliobaitė et al., 2016). Concept drift refers to the notion that users may change their interests over time and are unlikely to maintain static preferences.

**Reward shaping.** Similarly to the reward perturbation component, reward shaping modifies the reward. However, unlike the perturbed reward which is added to the memory, the reward modified by the reward shaping component is returned directly to the RL model and is not stored in memory. The reward shaping module aims to reflect changes in the reward that are not related to a change in the preference of a user, such as spontaneous decisions or fleeting interests.

## 4 EXPERIMENTS

To evaluate SUBER, we followed the approach of Rohde et al. (2018) and Ie et al. (2019a). We perform sanity checks and case studies, which we present in Section 4.2 and Section 4.3. To achieve this, we implemented a movie recommendation and a book recommendation environment in our framework. In the following sections, we discuss our implementation and design choices for these environments, as well as our ablation study and experiments. For the movie setting, we use rewards from 1 to 10, similar to TMDB, while for the book setting we use rewards from 1 to 5, as found in the Amazon Reviews Dataset (Ni et al., 2019).

For both environments, we created a dataset of synthetic users using Vicuna (Zheng et al., 2023) with Guidance (Lundberg, 2023). To generate user descriptions, we condition the LLM with information such as the age, liked and disliked genres, hobbies, and profession of the user (cf. Listing 1). We generate the user age by sampling from the age distribution in the United States (Bureau, 2022), while hobbies and professions were sampled from a precompiled list (cf. Appendix F).

For the movie dataset we use MovieLens (Harper & Konstan, 2015). For the book dataset, we used a subset of the Amazon Book Dataset (Ni et al., 2019). For more details, see Appendix A and Appendix B.

Listing 1: User generation and characteristic assignment process example by Vicuna with guidance. Black text shows the template and the instruction, red marks the sampled information from external distributions, blue indicates the content generated by LLM.

> *Can you generate details for a person, you need to generate a name, an age, a hobby, a job and a detailed, long and original description that contains the persons interests and secondary hobbies. Please outline the cinematic preferences of the individual, detailing their favorite and least favorite genres. Kindly provide explanations for each genre preference.*
> *Name: Emily Johnson, Age: 37, Gender: F, Hobby: compact discs, Job: detective*
> *Genres liked: romance, horror, Genres diskliked: action, comedy*
> *Description: she is a detective, she has a hobby of collecting compact discs, she likes to watch romance and horror movies in her free time, she dislikes action and comedy movies because she find them too chaotic and not interesting, her secondary hobbies are reading mystery novels and playing the piano.*

### 4.1 SETUP

We implemented three different approaches for the retrieval component: feature retrieval, recency retrieval, and similarity retrieval. The feature-based approach retrieves items based on the Sorensen Coefficient of movie genres, actors, director, and average rating, while for books, we use book category, author, and average rating. The recency algorithm retrieves the most recent interactions. The similarity approach retrieves items from the history based on their similarity to the query item. We generate item-description embeddings using a Sentence-T5 model (Ni et al., 2022) and compute their similarities based on the cosine distance. To select the item-rating pair to retrieve from memory, we compute the similarity between the query item and all items previously viewed by the current user, selecting the items with the highest similarity.

We use the items returned from the retrieval component to construct a prompt to query the LLM. The LLM is tasked with generating a rating of the queried item by the current user, where the queried item corresponds to the item suggested by the recommender system. We construct the prompt such that the user description comes first, allowing us to leverage the key-value cache (Pope et al., 2023), eliminating the need to recalculate all intermediate embeddings within the layers of the LLM for already encountered prefixes, therefore, increasing execution speed.

Furthermore, we experimented with one-shot and two-shot prompting to improve model performance, which has been shown to increase generation quality (Brown et al., 2020). In addition to the

default system prompt, we created a custom system prompt. Refer to Appendix A and Appendix B for details.

Tokenization ambiguity can become an issue when generating numbers with LLMs. Since we are dealing with ratings on a scale from one to ten, and because the number "10" can be tokenized in two different ways, this can cause unwanted side effects. To tackle this challenge, we tested two additional strategies for the movie setting: shifting all rewards to the scale of 0-9, and using words for numbers from "one" to "ten."

We experimented with various quantized versions of Llama and Vicuna, using LLMs that could run within a 24GB memory limit. A list of the models used in our experiments can be found in Appendix D. All models were quantized using GPTQ. Since different LLMs influence the simulation of human behavior in different ways. It is important to highlight the inherent trade-off between model size and processing speed. In particular, during training of an RL model, a fast environment is desirable to acquire more samples in a shorter time span. However, smaller LLMs may not adequately emulate the desired human behavior of our synthetic users.

For the reward perturbation experiment, we compared Gaussian noise and greedy noise. Greedy noise alters the LLM rating by 1 with a probability of $q$, while it remains unchanged with a probability of $1 - q$.

Our implementation of reward shaping operates on the following premise: As a user engages with an item more frequently, their interest in revisiting it diminishes. Conversely, as time passes, the likelihood of the user interacting with the item increases again (Russell & Levy, 2011). Given this insight, let us consider a user $u$ from the set $U$ and an item $i$ with which the user has interacted $n_{ui}$ times. When a time span of $\Delta t$ has passed since the last interaction with the item, the reward $r$ undergoes a reshaping process, characterized by the following equation:

$$r \leftarrow \max(1, \lfloor r \cdot q^{n_{ui}/\Delta t} \rfloor), \tag{2}$$

where $q \in [0, 1]$. This adjustment takes into account both the frequency of user interaction with the item and the time elapsed since their last interaction, resulting in the modified reward $r$.

## 4.2 ABLATIONS

To determine the effect of each component in our environment, we performed ablations across four different test cases. In this section we present the high-level idea, for more details, see Appendix C.

**Genres/Categories.** We assess the environment's ability to recognize movie and book genres, and its ability to correlate those genres with user preferences to accurately predict ratings. User profiles were created manually for each movie genre, ensuring that they express a preference for the selected genre while disliking all others. Afterwards, we queried the environment with users and movies from both their favored and disliked genres. The accuracy of rating predictions is used to measure performance. A similar process is used for the book environment, replacing movie genres with book categories.

**High/Low rating.** We assess whether the environment can accurately infer high ratings for users who provide positive-leaning descriptions, while inferring low ratings for users whose descriptions are negative-leaning. We give each user a set of items and test whether the environment is able to generate high or low ratings, depending on the description of the user.

**Collection of items.** We evaluate the ability of the environment to leverage the historical item ratings of a user to predict their future ratings. We conduct this test by manually selecting a set of item collections belonging to a series (e.g., James Bond, Toy Story, etc.). Subsequently, we randomly select a sample of users from our synthetic dataset and fill their history with items from our collection as well as random items. We assign a high rating to all items in the collection history, and the corresponding average rating to the remaining random items. Success is measured by a high rating for the queried item that is part of the collection. The experiment is repeated by assigning low ratings to the collection items to test the ability of the environment to predict low ratings.

**Similarity to real rating distribution.** We evaluate if the rating distribution obtained from our environment accurately reflects human behavior by comparing it to the rating distribution of MovieLens and Goodreads (Dimitrov et al., 2015), which are representative samples of human ratings. We sample with replacement from our environment as well as from MovieLens and Goodreads datasets. We

then calculate the empirical distribution across these datasets and utilize the total variation distance as a metric to measure similarity.

The aggregated score is the mean of all test cases. All ablations, except where defined otherwise, were performed using the following configurations. We used the 2-shot prompting, a custom system prompt, three item retrievial via T5-similarity, and no reward perturbation. For movies, we used *Vicuna-v.1.5-13B* with rating scale 0-9, and for books we use *Llama-2-Chat-13B* with scale 1-5.

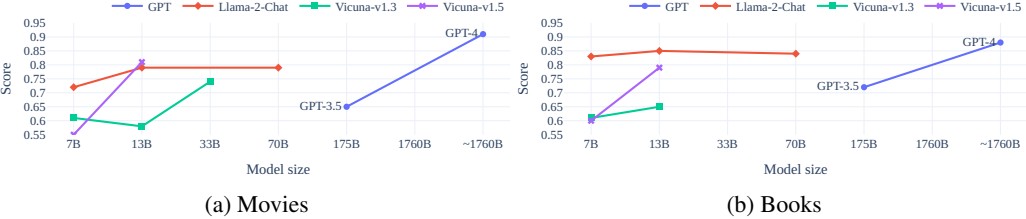

(a) Movies                    (b) Books

Figure 3: Aggregated score across LLM families for the movie environment (a), and for the book environment (b) by varying only the LLM component. For numerical values of the scores of each test case, see Appendix D.

Table 2: Comparison of different retrieval strategies on the *collection of items* test set for movies and books. We only show the entries of the test case where the result changed significantly, for extended results see Appendix D.

| Retrieval component | Collection of movies ↑ | Collection of books ↑ |
|---|---|---|
| None | 0.48±0.01 | 0.50±0.00 |
| Most recent | 0.68±0.01 | 0.64±0.00 |
| T5 similarity | 0.82±0.02 | 0.78±0.06 |
| Feature similarity | **0.83±0.02** | **0.82±0.07** |

Table 3: Comparison of different perturbation components on the *similarity to real rating distribution* test for movies and books.

| Perturbation component | Similarity to ML ↑ | Similarity to GR ↑ |
|---|---|---|
| None | 0.75±0.00 | 0.85±0.00 |
| Greedy | 0.79±0.00 | **0.87±0.00** |
| Gaussian | **0.82±0.00** | 0.86±0.00 |

**Results.** In the movie environment, we observe the impact of few-shot prompting and our custom prompt on performance (cf. Table 4). The best results emerge from the combination of two-shot prompting and our custom system prompt with a rating scale between 0 and 9, which has the best aggregated score and performance in three out of four test sets. In particular, using a rating scale between 1 and 10 performs poorly, even worse than zero-shot prompting with the default system prompt. Furthermore, prompts using word ratings, while not the best, show improvement compared to the numeric scale from 1 to 10. In Table 4, we observe that the best environment configuration achieves 82% accuracy for the movie collection test set, and 69% accuracy on the genres score,

Table 4: Ablation results for the movie setting using *Vicuna-v1.5-13B* as our environment. We test the LLM on coherency and realistic ratings for user-movie interactions. We achieve best performance with 0-9 digit rating scale, 2-shot prompting, and our custom system prompt.

| Prompt component | | | | | | | |
|---|---|---|---|---|---|---|---|
| Rating scale | N-shot | System prompt | Genres ↑ | High/Low ↑ | Collection of movies ↑ | Similarity to ML ↑ | Agg. score ↑ |
| 0-9 | 0-shot | default | 0.65±0.00 | 0.99±0.00 | 0.62±0.02 | 0.64±0.00 | 0.72±0.00 |
| 0-9 | 0-shot | custom | 0.69±0.00 | 0.99±0.00 | 0.64±0.02 | 0.65±0.00 | 0.74±0.00 |
| 0-9 | 1-shot | default | 0.61±0.00 | **1.00±0.00** | 0.71±0.01 | **0.75±0.00** | 0.77±0.00 |
| 0-9 | 1-shot | custom | **0.72±0.00** | **1.00±0.00** | 0.74±0.03 | 0.74±0.00 | 0.80±0.01 |
| 0-9 | 2-shot | default | 0.63±0.01 | **1.00±0.00** | 0.81±0.02 | 0.74±0.00 | 0.80±0.00 |
| 0-9 | 2-shot | custom | 0.69±0.00 | **1.00±0.00** | **0.82±0.02** | **0.75±0.00** | **0.81±0.00** |
| 1-10 | 2-shot | custom | 0.64±0.01 | 0.72±0.03 | 0.68±0.01 | 0.72±0.00 | 0.69±0.01 |
| one-ten | 2-shot | custom | 0.71±0.01 | **1.00±0.00** | 0.72±0.03 | 0.64±0.00 | 0.77±0.01 |

demonstrating the ability of the environment to capture human concepts such as genres and movie franchises. Similar trends are evident in the book environment (cf. Appendix D.2). In general, we observe that larger models perform better across model families (cf. Figure 3). Furthermore, a comparison between *Vicuna-1.5* and *Llama-2-Chat* reveals how fine-tuning the same foundation model (Llama-2) can influence performance.

Our ablation of the retrieval component demonstrates that this component plays a crucial role in understanding user interests (cf. Table 2). Furthermore, the recency approach proves inadequate, while the best-performing retrieval approach is predicated on the similarity of item features. We can also observe that the different perturbations slightly affect the similarity with the actual data distribution, especially in the case of the movie environment (cf. Table 3).

## 4.3 Benchmarks

We demonstrate the viability of our environment for training an RL recommender system. We implemented four different agents based on A2C (Mnih et al., 2016), PPO (Schulman et al., 2017), TRPO (Schulman et al., 2015), and DQN (Mnih et al., 2013). We train all models for 1.6M steps on SUBER. See Appendix E for more details.

Table 5: Results of RL methods trained on SUBER. "Liked genres" refers to the percentage of movies in the top ten recommendations that belong to the preferred genres of the user (cf. Appendix E). "Pers.@10" refers to the personalization over the first ten recommendations.

| Algorithm | Average reward | MAP@10 ↑ | MMR@10 ↑ | Pers.@10 ↑ | % Liked genres ↑ | % Disliked genres ↓ |
|---|---|---|---|---|---|---|
| DQN | 6.87±0.02 | 0.43±0.06 | 0.71±0.14 | 0.00±0.00 | 0.43±0.01 | 0.18±0.01 |
| PPO | 7.01±0.01 | 0.59±0.01 | 0.88±0.00 | **0.99±0.00** | 0.44±0.00 | 0.15±0.00 |
| TRPO | 7.19±0.02 | 0.63±0.05 | 0.84±0.10 | 0.41±0.02 | 0.46±0.01 | 0.14±0.01 |
| A2C | **7.77±0.04** | **0.88±0.01** | **0.96±0.01** | 0.85±0.03 | **0.51±0.00** | **0.10±0.01** |

We observe that A2C achieves the best overall performance in our case study (cf. Table 5). While PPO achieves a higher personalization score compared to A2C, PPO has more difficulty recommending items that are of interest to users. Additionally, we introduce two metrics to evaluate how well the agent recommends items which align to the preferences of users, where A2C also outperforms other tested approaches. Finally, we show two random qualitative examples of recommendations generated by A2C in Table 6.

Table 6: Random examples of movie recommendation from trained A2C model on SUBER.

| Watched movies by user | Movies recommended to user |
|---|---|
| Wallace & Gromit: The Curse of the Were-Rabbit, Mission: Impossible | Live Free or Die Hard, The World's Fastest Indian, The Fugitive |
| Jurassic Park, The Way of the Dragon, Master and Commander: The Far side of the World | Mission: Impossible, Batman Begins, The Bourne Supremacy |

## 5 Conclusion

Our research offers a possible avenue to address the persistent challenge of training recommender systems in the absence of real user interactions. Conventional approaches that depend on user-item interaction histories or synthetic data have often failed to replicate real-world usage scenarios accurately. By introducing SUBER, a novel RL environment designed specifically for recommender system training, and incorporating recent advances in LLMs to emulate human behavior in the training environment, we have proposed a potential solution to this long-standing issue. Our results, as demonstrated through a series of ablation studies and experiments, underscore the efficacy of our approach. We believe that this work marks a step toward achieving more realistic and practical training environments for recommender systems, even when direct user interactions are unavailable.

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
