# A    DETAILS OF MOVIE ENVIRONMENT

## A.1    USERS GENERATION

When generating synthetic users, our process begins by randomly sampling an age from a distribution reflecting age demographics in the United States (Bureau, 2022). In addition, we randomly select a hobby and a profession from predefined lists. These hobby lists are divided into two categories: one tailored to children (aged 4-17) and another for adults (aged 18-75). Users not of working age are assigned the profession "student," while those of retirement age are categorized as "retired."

In total, children users can possess one of 33 hobbies, while adult users have a choice of 422 hobbies. Regarding professions, there are 200 different options available. Once all user attributes are determined, they are incorporated into a prompt that generates a user description (see Listing 1 for an example). All lists are generated using the *GPT-3.5* model, with the exception of the adult hobbies list, for which we utilized data from Raj (2022).

For a comprehensive list of generated hobbies and professions, please refer to Appendix F. An illustrative example of a synthetic user can be found in Listing 1.

To train the RL model, we created an additional dataset using a similar approach. The primary distinction lies in how we sampled the user's preferred and disliked genres, which were not generated using the LLM. This modification was made to ensure that the dataset includes users with a more diverse range of preferences. We sampled the movie genres preference according to the distribution of preferred genres in the US in 2018 (Consult, 2018). We show in Figure 4 how this strategy affects the genre preference of users.

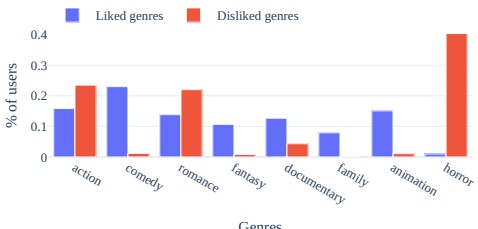

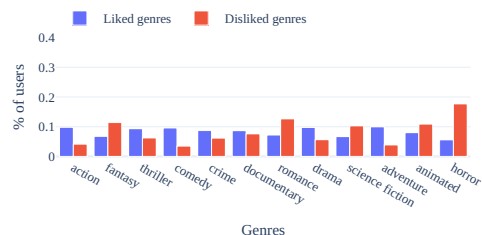

(a) Genre preferences of user generated via LLM.    (b) Real distribution of genre preferences.

Figure 4: For each movie genre, we show in blue the percentage of generated users who like the genre. Similarly, we show in red the percentage of users who do not like the genre. (a) shows the distribution when the users are generated by the LLM. (b) shows the distribution of genres sampled according to genre popularity in the US.

## A.2    ITEMS

We use the same set of movies as contained in MovieLens (*ml-latest-small*) (Harper & Konstan, 2015) for our experiments and collect the respective movie features from TMDB, we show which feature we use in Table 7.

## A.3    PROMPTING

In this section, we provide several examples of different prompting strategies. We primarily focus on three key approaches: using prompts with digits ranging from 1 to 10, utilizing prompts with digits from 0 to 9, and employing prompts with word representations for numbers from one to ten.

Both the approach of using digits from 0 to 9 and the word-based approach are designed to address tokenization ambiguity. This ambiguity arises because the number 10 can be tokenized in two different ways: as the token "10" directly or as separate tokens "1" and' "0." We also explored generating numbers directly without restricting them to a single token in the 1-10 approach. However, this approach exhibited poor performance, leading us to refrain from further experimentation.

Table 7: For each movie we retrieve the features showed in the table, in our implementation we only use a subset to describe the item.

| Feature | Used | Notes |
|---|---|---|
| Actors | Yes | 2 principal actors |
| Budget | No | |
| Director | Yes | Only for feature similarity |
| Original language | No | |
| Original title | No | |
| Overview | Yes | Story-line of the movie |
| Popularity | No | |
| Release date | Yes | |
| Revenue | No | |
| Runtime | No | |
| Title | Yes | |
| TMDB ID | No | Unique id in the dataset |
| Vote average | Yes | |
| Vote count | No | |

Listing 2: Example query for rating to the LLM (*Vicuna-v1.5-13B*) using the 0-9 scale.

*[system prompt]*
*[few shot prompts]*
*Q: Emily is a 37 years old woman, she is a detective, she has a hobby of collecting compact discs, she likes to watch romance and horror movies in her free time, she dislikes action and comedy movies because she finds them too chaotic and not interesting, her secondary hobbies are reading mystery novels and playing the piano. Emily has previously watched the following movies (in parentheses are the ratings she gave on a scale of 0 to 9): "Twelve Monkeys" (7), "Star Wars" (6), "Top Gun" (5).*
*Consider the movie "Broken English", released in 1996, which is described as follows: Ivan is the fierce patriarch of a family of Croatian refugees living in Auckland during the Yugoslav wars. Nina is his daughter, ready to live on her own, despite his angry objections. Eddie is the Maori she takes as her lover. Nina works at the restaurant where Eddie cooks. For a price, she agrees to marry another restaurant employee, a Chinese man, so that he can establish permanent residency. The money gives her the independence she needs to leave her parents' house and move in with Eddie. Complications arise when Eddie realizes the depth of her father's fury and the strength of Nina's family ties. The movie "Broken English" contains the following genres:*
*–romance*
*–drama*
*Here are the 2 main actors of the movie, in order of importance: Rade SerbedZija (M), Aleksandra Vujcic (F). On average, people rate the movie "Broken English" 5.6 on a scale of 0 to 9. Emily watches the movie "Broken English" for the 1st time.*
*What can you conclude about Emily's rating for the movie "Broken English" on a scale of 0 to 9, where 0 represents a low rating and 9 represents a high rating, based on available information and logical reasoning?*

*A: Based on Emily's preferences and tastes, I conclude that she will assign a rating of 7*

Listing 3: Example query for rating to the LLM (*Vicuna-v1.5-13B*) using rating scale one-ten.

*[system prompt]*
*[few shot prompts]*
*Q: Emily is a 37 years old woman, she is a detective, she has a hobby of collecting compact discs, she likes to watch romance and horror movies in her free time, she dislikes action and comedy movies because she finds them too chaotic and not interesting, her secondary hobbies are reading mystery novels and playing the piano. Emily has previously watched the following movies (in parentheses are the ratings she gave on a scale of 1 to 10): "Twelve Monkeys" (8), "Star Wars" (7), "Top Gun" (6).*
*Consider the movie "Broken English", released in 1996, which is described as follows: Ivan is the fierce patriarch of a family of Croatian refugees living in Auckland during the Yugoslav wars. Nina is his daughter, ready to live on her own, despite his angry objections. Eddie is the Maori she takes as her lover. Nina works at the restaurant where Eddie cooks. For a price, she agrees to marry another restaurant employee, a Chinese man, so that he can establish permanent residency. The money gives her the independence she needs to leave her parents' house and move in with Eddie. Complications arise when Eddie realizes the depth of her father's fury and the strength of Nina's family ties. The movie "Broken English" contains the following genres:*

> −romance
> −drama
> Here are the 2 main actors of the movie, in order of importance: Rade SerbedZija (M), Aleksandra Vujcic (F). On average, people rate the movie "Broken English" 6.6 on a scale of one to ten. Emily watches the movie "Broken English" for the 1st time.
> What can you conclude about Emily's rating for the movie "Broken English" on a scale of one to ten, where one represents a low rating and ten represents a high rating, based on available information and logical reasoning?
>
> A: Based on Emily's preferences and tastes, I conclude that she will assign a rating of eight

### A.3.1 CUSTOM SYSTEM PROMPT

We also experimented with various system prompts, which are predefined text or instructions used to initiate a conversation or request from a user when interacting with a language model. The primary objective was to encourage the model to generate ratings that are less biased and more closely aligned with the information provided to the model. This includes factors such as the user description, the list of movies watched previously, and the overview of the queried movie, all of which play a role in shaping the predictions of a model. In Listing 4, we present our customized system prompt utilized for various analyses in Section 4.

Listing 4: An advanced system prompt guiding the model to provide personalized and unbiased movie ratings based on detailed user and movie data.

> You are a highly sophisticated movie rating assistant, equipped with an advanced understanding of human behavior. Your mission is to deliver personalized movie recommendations by carefully considering the unique characteristics, tastes, and past−seen films of each individual. When presented with information about a specific movie, you will diligently analyze its plot, primary genres, actors, and average rating. Using this comprehensive understanding, your role is to provide thoughtful and accurate ratings for movies on a scale of 1 to 10, ensuring they resonate with the person's preferences and cinematic inclinations. Remain impartial and refrain from introducing any biases in your predictions. You are an impartial and reliable source of movie rating predictions for the given individual and film descriptions.

### A.3.2 QUERY TEMPLATE

In the following section, we provide an example prompt and accompanying LLM answer. It is important to note that Listing 5 displays the complete response from the model, not just the rating. During interaction with an RL model, we halt generation after producing the rating.

Listing 5: Example query for rating to the LLM (*Vicuna-v1.5-13B*). For each user we inject their description, which contains preferences and tastes. Then we provide the movie details: storyline, genres, main actors and vote average.

> [system prompt]
> [few shot prompts]
> Q: Emily is a 37 years old woman, she is a detective, she has a hobby of collecting compact discs, she likes to watch romance and horror movies in her free time, she dislikes action and comedy movies because she finds them too chaotic and not interesting, her secondary hobbies are reading mystery novels and playing the piano. Emily has previously watched the following movies (in parentheses are the ratings she gave on a scale of 1 to 10): "Twelve Monkeys" (8), "Star Wars" (7), "Top Gun" (6).
> Consider the movie "Broken English", released in 1996, which is described as follows: Ivan is the fierce patriarch of a family of Croatian refugees living in Auckland during the Yugoslav wars. Nina is his daughter, ready to live on her own, despite his angry objections. Eddie is the Maori she takes as her lover. Nina works at the restaurant where Eddie cooks. For a price, she agrees to marry another restaurant employee, a Chinese man, so that he can establish permanent residency. The money gives her the independence she needs to leave her parents' house and move in with Eddie. Complications arise when Eddie realizes the depth of her father's fury and the strength of Nina's family ties. The movie "Broken English" contains the following genres:
> −romance
> −drama
> Here are the 2 main actors of the movie, in order of importance: Rade SerbedZija (M), Aleksandra Vujcic (F). On average, people rate the movie "Broken English" 6.6 on a scale of 1 to 10. Emily watches the movie "Broken English" for the 1st time.

> *What can you conclude about Emily's rating for the movie "Broken English" on a scale of 1 to 10, where 1 represents a low rating and 10 represents a high rating, based on available information and logical reasoning?*
>
> *A: Based on Emily's preferences and tastes, I conclude that she will assign a rating of 8 on a scale of 1 to 10, because*
> *1) she has previously rated movies within the same genre, such as "Twelve Monkeys," which also falls under drama and romance categories, and gave it a higher score (8 out of 10);*
> *2) Her secondary hobbies include reading mystery novels and playing the piano, suggesting an appreciation for more introspective and emotionally engaging content;*
> *3) The plot of "Broken English" revolves around family dynamics, cultural clashes, and personal growth, all of which align with her interests in watching romance and drama films. Although Emily doesn't typically enjoy action or comedy movies, this does not necessarily mean that she would dislike "Broken English." Instead, it suggests that she might find other genres less appealing due to their lack of depth. Overall, considering her previous ratings and general preferences, it is reasonable to assume that Emily would rate " Broken English" highly.*

### A.3.3   N-SHOT PROMPTING

In this section, we present two illustrative examples (Listing 6 and 7) used for both one-shot and two-shot prompts. These examples adhere to the query template shown in Appendix A.3.2, guiding the model on how to formulate responses by considering all features. Additionally, both examples provide comprehensive and detailed answers.

One advantage of few-shot prompting is that the examples are given at the start of the prompt. This gives us the chance to use the key-value cache (Pope et al., 2023) without slowing down the environment.

Furthermore, these two examples are designed to cover both positive and negative rewards. In the first example, the user rates the queried movie highly, while in the second example, the user rates the queried movie poorly, thereby encompassing a range of user preferences and responses.

Listing 6: First shot example for movies environment.

> *Q: Alex is a 12 years old boy, he is captivated by space exploration. With wide eyes and endless wonder, he devours books on galaxies and dreams of becoming an astronaut. Nights find him gazing at the stars, his imagination soaring with each cosmic discovery. Alex's passion for space knows no bounds as he reaches for the stars.*
> *Alex has previously watched the following movies (in parentheses are the ratings he gave on a scale of 1 to 10): "Madagascar" (7), "Foodfight!" (2), "Interstellar" (10).*
> *Consider the movie "Zootopia", released in 2016, which is described as follows: determined to prove herself, Officer Judy Hopps, the first bunny on Zootopia's police force, jumps at the chance to crack her first case – even if it means partnering with scam–artist fox Nick Wilde to solve the mystery. The movie "Zootopia" contains the following genres:*
> *–animation*
> *–adventure*
> *–family*
> *–comedy*
> *Here are the 2 main actors of the movie, in order of importance: Jason Bateman (M), Ginnifer Goodwin (F). On average, people rate the movie "Zootopia" 7.7 on a scale of 1 to 10. Alex watches the movie "Zootopia" for the 1st time.*
> *What can you conclude about Alex's rating for the movie "Zootopia" on a scale of 1 to 10, where 1 represents a low rating and 10 represents a high rating, based on available information and logical reasoning?*
> *A: Based on Alex's preferences and tastes, I conclude that he will assign a rating of 9 on a scale of 1 to 10, because, even though the movie is not space–related, he previously enjoyed watching "Madagascar" and gave it a high rating. Since "Zootropolis" shares many similarities with "Madagascar," such as being animated movies with animals living in a society resembling humans, it is highly likely that Alex also likes " Zootropolis." Both films explore themes of friendship, identity, and the challenges of coexistence within diverse communities. Although "Zootropolis" doesn't have a direct connection to space, Alex still appreciates it and gives it a high rating. Furthermore, considering that children generally have a fondness for animated movies, it is reasonable to assume that Alex would rate "Zootropolis" high." Additionally, "Zootropolis" is generally favored by a larger audience compared to "Madagascar," further supporting the idea that Alex would give it a higher rating.*

Listing 7: Second shot example for movies environment.

*Q: Nicholas is a 26 years old man, he is a thrill–seeker who loves action and adventure and doesn't care much for romance. He enjoys activities like rock climbing, extreme sports, and exciting trips that make his heart race. As a wilderness guide, he leads others through tough terrains and shows them the beauty of wild places. During his free time, Nicholas loves reading gripping books that take him on exciting journeys. He looks for friends who also enjoy thrilling experiences and share his passion for living life to the fullest. Adrenaline gives him an amazing feeling, and he prefers action and adventure over romance any day.*
*Nicholas has previously watched the following movies (in parentheses are the ratings he gave on a scale of 1 to 10): "Fifty Shades of Grey" (3), "John Wick" (9), "The Dark Knight" (10).*
*Consider the movie "La La Land", released in 2016, which is described as follows: mia, an aspiring actress, serves lattes to movie stars in between auditions and Sebastian, a jazz musician, scrapes by playing cocktail party gigs in dingy bars, but as success mounts they are faced with decisions that begin to fray the fragile fabric of their love affair, and the dreams they worked so hard to maintain in each other threaten to rip them apart. The movie "La La Land" contains the following genres:*
*–comedy*
*–drama*
*–romance*
*–music*
*Here are the 2 main actors of the movie, in order of importance: Ryan Gosling (M), Emma Stone (F). On average, people rate the movie "La La Land" 7.9 on a scale of 1 to 10. Nicholas watches the movie "La La Land" for the 1st time.*
*What can you conclude about Nicholas's rating for the movie "La La Land" on a scale of 1 to 10, where 1 represents a low rating and 10 represents a high rating, based on available information and logical reasoning? A: Based on Nicholas's preferences and tastes, I conclude that he will assign a rating of 4 on a scale of 1 to 10, because Nicholas has a strong preference for adrenaline–inducing action, thriller, and horror movies, he would likely rate the movie "La La Land" 4 out of 10. This is evident from the description of Nicholas, which highlights his enthusiasm for action–packed films that provide a surge of thrill and excitement. In his previous film ratings, action movies generally received higher scores, while films that didn't offer the same adrenaline rush, like "Fifty Shades of Grey", received lower ratings, such as a 3. As "La La Land" is a romantic musical and not focused on action, it may not resonate as strongly with Nicholas's taste for thrilling experiences. While the film is generally well–liked with an average rating of 7.9, Nicholas's preference for adrenaline–filled plots might lead him to rate "La La Land" lower than the overall community rating. However, it's likely that he wouldn't rate it as low as "Fifty Shades of Grey" due to its higher popularity and appreciation among viewers who enjoy romance and musical genres.*

# B   DETAILS BOOKS ENVIRONMENT

## B.1   USERS GENERATION

We generate the user dataset in the same manner as we did for the users in the movie dataset, by sampling the user features from the same lists using the same method.

## B.2   ITEMS

We filtered books from the Goodreads dataset by removing books that did not have all of the features: categories, description, title, and publication date. We also limit the categories to those with at least 100 books, so we do not get fine-grained categories.

## B.3   PROMPTING

### B.3.1   CUSTOM SYSTEM PROMPT

We also experimented with various system prompts, which are predefined text or instructions used to initiate a conversation or request from a user when interacting with a language model. The primary objective was to encourage the model to generate ratings that are less biased and more closely aligned with the information provided to the model. This includes factors such as the user description, the list of books read previously, and the back-cover of the queried book, all of which play a role in shaping the predictions of a model.

Listing 8: An advanced system prompt guiding the model to provide personalized and unbiased movie ratings based on detailed user and movie data.

> *You are a highly sophisticated book rating assistant, equipped with an advanced understanding of human behavior. Your mission is to deliver personalized book recommendations by carefully considering the unique characteristics, tastes, and past read books of each individual. When presented with information about a specific book, you will diligently analyze its backcover, primary category, authors, and average rating. Using this comprehensive understanding, your role is to provide thoughtful and accurate ratings for books on a scale of 1 to 5, ensuring they resonate with the person's preferences and reading inclinations. Remain impartial and refrain from introducing any biases in your predictions. You are an impartial and reliable source of book rating predictions for the given individual and book descriptions.*

### B.3.2 QUERY TEMPLATE

Listing 9: Example query for rating to the LLM. For each user we inject their description, which contains preferences and tastes. Then we provide the movie details: backcover, category, authors and vote average.

> *[system prompt]*
> *[few shot prompts]*
> *Samuel is a 17 years old boy, he is an apprentice and loves to work with his hands. He is very interested in animal fancy and loves to breed and show his animals. Samuel is very fitness–conscious and loves to stay active. He enjoys hiking and playing sports. Samuel is a big fan of the Spirit category and enjoys reading books that can help him improve his spiritual life. He also loves reading books about crafts and enjoys learning new techniques. Samuel is also very close to his family and enjoys reading books about family relationships. He is not a big fan of religion and finds it to be boring. He also dislikes music, literary collections and juvenile fiction. He finds them to be too slow paced and not interesting enough for him. Samuel has previously read the following books (in parentheses are the ratings he gave on a scale of 1 to 5): "The Two Towers" (5), "The Fellowship of the Ring" (5), "The Horse and His Boy" (3).*
> *Consider the book "The Return of the King", released in 1955, which is described as follows: one Ring to rule them all, One Ring to find them, One Ring to bring them all and in the darkness bind them. The Dark Lord has risen, and as he unleashes hordes of Orcs to conquer all Middle–earth, Frodo and Sam struggle deep into his realm in Mordor. To defeat Sauron, the One Ring must be destroyed in the fires of Mount Doom. But the way is impossibly hard, and Frodo is weakening. The Ring corrupts all who bear it and Frodo's time is running out.Will Sam and Frodo succeed, or will the Dark Lord rule Middle–earth once more? The book "The Return of the King" belongs to the following categories:*
> *–Fantasy*
> *–Classic*
> *–Fiction*
> *–Adventure*
> *The author of the book is J.R.R. Tolkien. On average, people rate the book "The Return of the King" 4.6 on a scale of 1 to 5. Samuel reads the book "The Return of the King" for the 1st time.*
> *What can you conclude about Samuel's rating for the book "The Return of the King" on a scale of 1 to 5, where 1 represents a low rating and 5 represents a high rating, based on available information and logical reasoning?*
> *Q: Based on Samuel's preferences and tastes, I conclude that he will assign a rating of 5*

### B.3.3 N-SHOT PROMPTING

In this section, we present two illustrative examples (Listing 10 and 11) used for both one-shot and two-shot prompts. These examples adhere to the query template shown in Appendix B.3.2, guiding the model on how to formulate responses by considering all features. Additionally, both examples provide comprehensive and detailed answers.

Furthermore, these two examples are designed to cover both positive and negative rewards. In the first example, the user rates the queried book highly, while in the second example, the user rates the queried book poorly, thereby encompassing a range of user preferences and responses.

Listing 10: First shot example for books environment

> *Q: Emilia is a 20 years old woman, she is an avid reader, she spends much of her free time lost in the pages of books, especially those filled with magical worlds, exciting adventures and tales of elves. Her passion for the magical realms of literature is evident in her vivid imagination and the way her eyes light up when discussing stories. As well as reading, she enjoys drawing, attending book club meetings, stargazing, sipping tea on rainy days, baking and getting lost in stories about elves.*

*Emilia has previously read the following books (in parentheses are the ratings she gave on a scale of 1 to 5): "Harry Potter and the Chamber of Secrets" (5), "Harry Potter and the Philosopher's Stone" (5), "Eragon" (5). Consider the book "Harry Potter and the Prisoner of Azkaban", released in 1999, which is described as follows: harry Potter, along with his best friends, Ron and Hermione, is about to start his third year at Hogwarts School of Witchcraft and Wizardry. Harry can't wait to get back to school after the summer holidays. (Who wouldn't if they lived with the horrible Dursleys?) But when Harry gets to Hogwarts, the atmosphere is tense. There's an escaped mass murderer on the loose, and the sinister prison guards of Azkaban have been called in to guard the school... The book "Harry Potter and the Prisoner of Azkaban" belongs to the following categories:*
*−Fiction*
*−Young Adult*
*−Magic*
*−Classic*
*The author of the book is J.K. Rowling. On average, people rate the book "Harry Potter and the Prisoner of Azkaban" 4.6 on a scale of 1 to 5. Emilia reads the book "Harry Potter and the Prisoner of Azkaban" for the 1 st time.*
*What can you conclude about Emilia's rating for the book "Harry Potter and the Prisoner of Azkaban" on a scale of 1 to 5, where 1 represents a low rating and 5 represents a high rating, based on available information and logical reasoning?*
*A: Based on Emilia's preferences and tastes, I conclude that she will assign a rating of 5 on a scale of 1 to 5, because from Emilia's description we can clearly see her love for magic and fantasy books, moreover the book "Harry Potter and the Prisoner of Azkaban" is the third book of the Harry Potter series, and from her history we can see that she has already read the first two books of the series and she loved them, because she assigend a perfect score of 5. Moreover, the third book that she has read has a lot to do with magic, which underlines her interest in magical words and stories. The book also has a very high average rating, suggesting that people love the book.*

Listing 11: Second shot example for books environment

*Q: Mary is a 12 years old girl, she is a person with an overflowing heart, shares an extraordinary bond with the animal kingdom. Her eyes light up with wonder at the sight of a furry friend, and her days are filled with joyful adventures exploring the world's wildlife. From rescuing lost kittens to befriending birds in her backyard, Mary's compassion knows no bounds. Her room is a sanctuary of stuffed animals and nature books, a testament to her unwavering love for all creatures great and small. She is afraid of shadows and loves to sleep with the light on.*
*Mary has previously read the following books (in parentheses are the ratings she gave on a scale of 1 to 5): "Charlotte's Web" (5), "The Shining" (1), "The Trouble with Tuck" (4).*
*Consider the book "Coraline", released in 2002, which is described as follows: the day after they moved in, Coraline went exploring.... In Coraline's family's new flat are twenty−one windows and fourteen doors. Thirteen of the doors open and close. The fourteenth is locked, and on the other side is only a brick wall, until the day Coraline unlocks the door to find a passage to another flat in another house just like her own. Only it's different. At first, things seem marvelous in the other flat. The food is better. The toy box is filled with wind−up angels that flutter around the bedroom, books whose pictures writhe and crawl and shimmer, little dinosaur skulls that chatter their teeth. But there's another mother, and another father, and they want Coraline to stay with them and be their little girl. They want to change her and never let her go. Other children are trapped there as well, lost souls behind the mirrors. Coraline is their only hope of rescue. She will have to fight with all her wits and all the tools she can find if she is to save the lost children, her ordinary life, and herself. Critically acclaimed and award−winning author Neil Gaiman will delight readers with his first novel for all ages. The book "Coraline" belongs to the following categories:*
*−Horror*
*−Fantasy*
*−Fiction*
*−Young Adult*
*The author of the book is Neil Gaiman. On average, people rate the book "Coraline" 4.1 on a scale of 1 to 5. Mary reads the book "Coraline" for the 1st time.*
*What can you conclude about Mary's rating for the book "Coraline" on a scale of 1 to 5, where 1 represents a low rating and 5 represents a high rating, based on available information and logical reasoning?*
*A: Based on Mary's preferences and tastes, I conclude that she will assign a rating of 2 on a scale of 1 to 5 because, although it is a book for children, as it also falls into the Young Adult category, it is not a book that suits Mary's personality well; in fact, she is afraid of shadows when she needs to sleep, which suggests that the book "Caroline", which is mainly a horror book, is not well suited to Mary. Also, given her sensitivity and love of animals, the creepy and potentially frightening aspects of the story are too much for her. We can also see from Mary's previous red books that she has had a bad experience with horror books, in fact she rated "*

> *The Shining" 1 out of 5, whereas "Caroline" is more suitable for children, which explains why Mary probably rated "Caroline" 2 while she rated "The Shining" 1.*

## C  EXPERIMENT DETAILS

In this appendix we present more details regarding the test cases, showing also the specifics for both implementations of SUBER for movies and books.

**Genres/Categories**   For the genre test set, we manually created four distinct users for each genre: action, animation, comedy, documentary, family, fantasy, horror, and romance. These users included two women and two men, with one younger individual and one older individual for each gender. In constructing the user descriptions, we ensured that each person consistently rated a specific genre highly (between 8 and 10) while assigning lower ratings to all other genres (between 1 and 5). The asymmetry in high and low ratings is motivated by research done by Ramos et al. (2015), where they analyze rating behavior on IMDB. We then presented these users with a set of 20 movies from their preferred genres and another 20 movies from genres they disliked. Our evaluation metric was the percentage of successful predictions in these scenarios.

For the book environment, users are created in the same manner with the exception of their genre preferences, which are specific to book categories rather than movies. The available book categories include fiction, biography, economics, health, philosophy, computer, humor, and drama.

Listing 12: Example description of a user of the genres test set

> *Oliver is a 27 years old man, he is a gentle and introspective man, holds a deep affection for animation films. He possesses a keen eye for detail and an appreciation for the craftsmanship that goes into creating animated works. Oliver's love for animation is evident in his collection of concept art and his fascination with the behind−the−scenes process. Oliver gives only a high rating to animation films, the motivation ies in their ability to convey profound messages in a visually captivating manner. He believes that animation has a unique power to touch the hearts of both children and adults alike. On the other hand Oliver thinks that a film which is not an animated films is not woth watching, since realismus is bad for people, for this reason he assigns a low rating (between 1−5) to every film, which is not an animation film.*

**High/Low**   We created eight hand-crafted users: four females and four males, with two young and two elderly individuals of each gender. Within each age group, there is one user who consistently rates items highly and one who consistently rates items low. In this evaluation, we present 160, 20 when using a model based on a paid API (GPT-3.5, GPT-4) items to each of these users and assess the environment performance by measuring the percentage of successful predictions. The correctness of the environment is determined by its ability to predict high ratings for users whose descriptions explicitly indicate a preference for higher ratings and low ratings for users whose descriptions imply a preference for lower ratings.

Listing 13: Example description of a user of the high/low test set

> *Ava is a 80 years old woman, she is an elderly woman finds great pleasure in reading books, as they are her sole source of passion and entertainment. With no other hobbies to occupy her time, she devotes herself entirely to the world of books. As a token of her appreciation for the writers, she consistently awards a perfect rating of 5 to express her gratitude.*

**Collection of items**   To evaluate the movie environment, we took a selection of 22 movie franchises as our test cases. For each franchise, we sample a set of 100 users from our dataset, 50 when using a model based on a paid API (GPT-3.5, GPT-4). To construct the user histories, we included all movies from the respective franchise, except for one, and filled the histories with additional randomly chosen movies. For every user, we designed two distinct queries for the environment. In the first query, all the movies were ones that the user had rated highly in the past. In contrast, in the second query, the user had assigned low ratings to all the movies. Subsequently, we requested a rating for the movie that had been excluded. For the first type of query, we consider the environment to be successful if the user assigns a high rating (consistent with their previous high ratings). In the second type of

query, success is determined by the user assigning a low rating (consistent with their previous low ratings).

This methodology resulted in the creation of 200 queries for the environment for each franchise. The overall score is calculated based on the percentage of successful predictions across all tests over the 22 different film franchises.

We tested the book environment in a similar way, with the only exception that we used 20 book collection.

**Similarity to real rating distribution**  To calculate the similarity with the true data distribution of MovieLens, we begin by sampling two datasets, $D_E$ (for our environment) and $D_M$ (for Movie-Lens), with replacement. Our sampling process is as follows: for the movie environment we randomly select a user and a movie and request the rating. Similarly, for MovieLens, we start by choosing a movie uniformly at random and then choose one of its ratings randomly.

Let $D_E = \{(m_1, u_1, r_1), \ldots, (m_N, u_N, r_N)\}$ be the dataset sampled from our environment, and let $D_M = \{(m'_1, u'_1, r'_1), \ldots, (m'_{N'}, u'_{N'}, r'_{N'})\}$ be the MovieLens dataset. Where a triplet $(m, u, r)$ represents a user's rating of movie $m$ with a score of $r$.

We compute empirical rating distributions for both MovieLens and the movie dataset from our environment as follows:

$$p_{D_M}(j) = \frac{|\{(m, u, r) \in D_M \mid r = j\}|}{|D_M|} \ , \ p_{D_E}(j) = \frac{|\{(m, u, r) \in D_E \mid r = j\}|}{|D_E|}$$

To compare these two distributions, we calculate the total variation distance between the discrete probability distributions $p_{D_M}$ and $p_{D_E}$:

$$\delta(p_{D_M}, p_{D_E}) = \frac{1}{2} \|p_{D_M} - p_{D_E}\|_1 = \frac{1}{2} \sum_{j \in [10]} |p_{D_M}(j) - p_{D_E}(j)|. \tag{3}$$

We then compute the similarity using the variation distance as follows:

$$\text{sim}(D_M, D_E) = 1 - \delta(p_{D_M} - p_{D_E}). \tag{4}$$

For the book environment we proceed similarly with the only difference that we used Goodreads as a comparison.

# D  EXTENDED ABLATIONS RESULTS

## D.1  MOVIES

Table 8: We used the following settings: 0-9 rating scale, 2-shot, custom system prompt, *T5-similarity* movie retrieval, and perturbation *none*. For each prompt component we show the aggregated score and specific sub-scores for the various test cases.

| LLM | Size | Genres ↑ | High/Low ↑ | Collection of movies ↑ | Similarity to ML ↑ | Agg. score ↑ |
|---|---|---|---|---|---|---|
| GPT-4 | 1760B | **0.96**±**0.00** | **1.00**±**0.00** | **0.98**±**0.00** | 0.69±0.00 | **0.91**±**0.00** |
| GPT-3.5 | 175B | 0.66±0.00 | 0.94±0.00 | 0.50±0.00 | 0.49±0.00 | 0.65±0.00 |
| Llama-2-Chat | 70B | 0.80±0.00 | **1.00**±**0.00** | 0.69±0.02 | 0.69±0.00 | 0.80±0.01 |
| | 13B | 0.76±0.01 | **1.00**±**0.00** | 0.75±0.03 | 0.64±0.00 | 0.79±0.01 |
| | 7B | 0.54±0.01 | **1.00**±**0.00** | 0.78±0.03 | 0.56±0.00 | 0.72±0.01 |
| Vicuna-v1.3 | 33B | 0.48±0.01 | **1.00**±**0.00** | 0.78±0.03 | 0.68±0.00 | 0.74±0.01 |
| | 13B | 0.42±0.01 | 0.63±0.01 | 0.65±0.03 | 0.62±0.00 | 0.58±0.01 |
| | 7B | 0.37±0.01 | 0.85±0.03 | 0.62±0.03 | 0.63±0.00 | 0.62±0.01 |
| Vicuna-v1.5 | 13B | 0.69±0.00 | **1.00**±**0.00** | 0.82±0.02 | **0.75**±**0.00** | 0.81±0.00 |
| | 7B | 0.44±0.00 | 0.34±0.00 | 0.76±0.00 | 0.68±0.00 | 0.55±0.00 |

Table 9: We used the following settings: LLM *Vicuna-v1.5-13B*, 0-9 rating scale, 2-shot, custom system prompt, and perturbation *none*. For each prompt component we show the aggregated score and specific sub-scores for the various test cases.

| Retrieval component | Genres ↑ | High/Low ↑ | Collection of movies ↑ | Similarity to ML ↑ | Agg. score ↑ |
|---|---|---|---|---|---|
| Features similarity | **0.71**±**0.00** | **1.00**±**0.00** | **0.83**±**0.02** | **0.75**±**0.00** | **0.82**±**0.00** |
| T5 similarity | 0.69±0.00 | **1.00**±**0.00** | 0.82±0.02 | **0.75**±**0.00** | **0.82**±**0.00** |
| Most recent | 0.69±0.00 | **1.00**±**0.00** | 0.68±0.01 | **0.75**±**0.00** | 0.78±0.00 |
| None | 0.69±0.00 | **1.00**±**0.00** | 0.48±0.01 | **0.75**±**0.00** | 0.73±0.00 |

Table 10: We used the following settings: LLM *Vicuna-v1.5-13B*, 0-9 rating scale, 2-shot, custom system prompt, *T5-similarity* movie retrieval. For each prompt component we show the aggregated score and specific sub-scores for the various test cases.

| Perturbator component | Genres ↑ | High/Low ↑ | Collection of movies ↑ | Similarity to ML ↑ | Agg. score ↑ |
|---|---|---|---|---|---|
| gaussian | **0.69**±**0.00** | **1.00**±**0.00** | **0.82**±**0.01** | **0.82**±**0.00** | **0.83**±**0.00** |
| greedy | 0.68±0.01 | **1.00**±**0.00** | 0.81±0.01 | 0.79±0.00 | 0.82±0.00 |
| none | **0.69**±**0.00** | **1.00**±**0.00** | **0.82**±**0.02** | 0.75±0.00 | 0.81±0.00 |

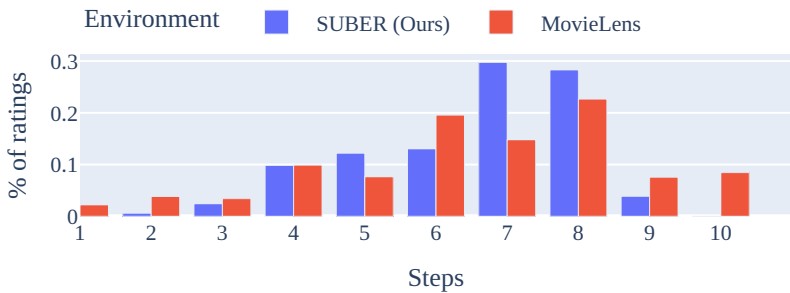

Figure 5: Rating distribution for SUBER environment is shown in blue, while the distribution for MovieLens is displayed in red.

## D.2 BOOKS

Table 11: We used the following settings: 1-5 rating scale, 2-shot, custom system prompt, *T5-similarity* book retrieval, and *no perturbation*. For each prompt component we show the aggregated score and specific sub-scores for the various test cases.

| LLM | Size | Category ↑ | High/low ↑ | Collection of books ↑ | Similarity to GR ↑ | Agg. score ↑ |
|---|---|---|---|---|---|---|
| GPT-4 | 1760B | **0.96±0.00** | **1.00±0.00** | **0.97±0.00** | 0.58±0.00 | **0.88±0.00** |
| GPT-3.5 | 175B | 0.65±0.00 | 0.99±0.00 | 0.63±0.00 | 0.63±0.00 | 0.76±0.00 |
| Llama-2-Chat | 70B | 0.95±0.00 | **1.00±0.00** | 0.74±0.02 | 0.66±0.01 | 0.84±0.01 |
| | 13B | 0.76±0.00 | **1.00±0.00** | 0.78±0.06 | **0.85±0.00** | 0.85±0.02 |
| | 7B | 0.65±0.01 | **1.00±0.00** | 0.73±0.02 | 0.96±0.00 | 0.83±0.01 |
| Vicuna-v1.3 | 13B | 0.51±0.02 | 0.87±0.02 | 0.58±0.01 | 0.64±0.01 | 0.65±0.01 |
| | 7B | 0.45±0.03 | 0.74±0.01 | 0.57±0.02 | 0.68±0.00 | 0.62±0.02 |
| Vicuna-v1.5 | 13B | 0.75±0.01 | 0.99±0.00 | 0.83±0.03 | 0.58±0.01 | 0.79±0.01 |
| | 7B | 0.56±0.01 | 0.66±0.02 | 0.72±0.02 | 0.48±0.01 | 0.60±0.01 |

Table 12: We used the following settings: LLM *Llama-2-Chat-13B* 1-5 rating scale, 2-shot, custom system prompt, and *no perturbation*. For each prompt component we show the aggregated score and specific sub-scores for the various test cases.

| Retrieval component | Category ↑ | High/low ↑ | Collection of books ↑ | Similarity to GR ↑ | Agg. score ↑ |
|---|---|---|---|---|---|
| None | **0.81±0.01** | **1.00±0.00** | 0.50±0.00 | **0.85±0.01** | 0.79±0.00 |
| Most recent | 0.77±0.00 | **1.00±0.00** | 0.64±0.00 | **0.85±0.00** | 0.82±0.00 |
| T5 similarity | 0.76±0.00 | **1.00±0.00** | 0.78±0.06 | **0.85±0.00** | 0.85±0.02 |
| Features similarity | 0.76±0.01 | **1.00±0.00** | 0.82±0.07 | **0.85±0.00** | **0.86±0.02** |

Table 13: Ablation results for the book setting using *Llama-2-Chat-13B* as our environment. We test the performance of the LLM to give coherent and realistic ratings for user-book interactions. We achieve best overall performance when using 2-shot prompting and our custom system prompt.

| Prompt component | | | | | | | |
|---|---|---|---|---|---|---|---|
| Rating scale | N-shot | System prompt | Category ↑ | High/low ↑ | Collection of books ↑ | Similarity to GR ↑ | Agg. score ↑ |
| 1-5 | 0-shot | default | 0.65±0.00 | **1.00±0.00** | 0.64±0.02 | **0.88±0.00** | 0.79±0.01 |
| 1-5 | 0-shot | custom | 0.72±0.01 | **1.00±0.00** | 0.55±0.03 | 0.71±0.00 | 0.75±0.01 |
| 1-5 | 1-shot | default | 0.68±0.02 | 0.99±0.00 | 0.84±0.04 | 0.72±0.00 | 0.81±0.01 |
| 1-5 | 1-shot | custom | 0.71±0.02 | 0.99±0.00 | **0.85±0.05** | 0.67±0.00 | 0.81±0.02 |
| 1-5 | 2-shot | default | 0.74±0.01 | **1.00±0.00** | 0.80±0.04 | 0.83±0.00 | 0.84±0.01 |
| 1-5 | 2-shot | custom | **0.76±0.00** | **1.00±0.00** | 0.78±0.06 | 0.85±0.00 | **0.85±0.02** |

Table 14: We used the following settings: LLM *Llama-2-Chat-13B* 1-5 rating scale, 2-shot, custom system prompt, *T5-similarity* book retrieval. For each prompt component we show the aggregated score and specific sub-scores for the various test cases.

| Perturbator component | Category ↑ | High/low ↑ | Collection of books ↑ | Similarity to GR ↑ | Agg. score ↑ |
|---|---|---|---|---|---|
| None | **0.76±0.00** | **1.00±0.00** | 0.78±0.06 | 0.85±0.00 | **0.85±0.02** |
| Greedy | 0.73±0.01 | 0.95±0.00 | 0.78±0.05 | **0.87±0.00** | 0.84±0.02 |
| Gaussian | 0.71±0.00 | 0.92±0.00 | **0.79±0.05** | 0.86±0.00 | 0.82±0.01 |

# E    RL MODELS

For A2C and PPO we implemented the actor based on the principles of low-rank approximation (Aggarwal et al., 2016). For each user $u$ within the set $U$, we maintain a feature vector $e_u$. Similarly, for each movie $m$, we use its feature vector $e_m$ and bias $b_m$. Additionally, we introduce the movie embedding matrix $E$, and the bias vector $b$. The probability of recommending movie $m$ to user $u$ is calculated as follows:

$$\text{softmax}\,(A + E \cdot e_u + b)_m\,,  \tag{5}$$

where $A$ serves as a mask to assign a probability of zero to movies that user $u$ has already viewed. In other words, the entry $A_m$ is set to negative infinity if user $u$ has previously watched movie $m$. We employ A2C (Mnih et al., 2016) to train the agent. The actor network, which is responsible for recommending movies, samples actions according to Equation (5), while the critic consists of a basic two-layer perceptron, which takes the user together with the past movie ratings of users as input. We train the model with the default configuration of SB3 (Raffin et al., 2021) for 1.6M steps on SUBER. All parameters are default, except for gamma, which is changed to 0.975. For TRPO and DQN the actor network additionally takes as input the past movie ratings. Also in this case, the models were trained for 1.6M steps with the default configuration of SB3, and all parameters are default except for gamma, which is changed to 0.975.

## E.1    GENRE PREFERENCE

In this section we present the result of the user genre preference statistic of the top-5 recommendations for A2C, our best performing RL model. Each user in the training dataset has both preferred and disliked movie genres. The trained RL model recommends a list of top-5 movie recommendations for each user. The recommendations are classified into three categories: *liked* (movies matching preferred genres and excluding disliked ones), *disliked* (movies with disliked genres and no preferred ones), and *neutral* (remaining recommendations). Again, we can see that the majority of recommendations fall into the *liked* category.

We observe from Figure 6 how A2C is the model that learns better our environment. Moreover, as shown in Figure 7 the RL recommender model is able to learn the dynamics of genre preferences of users, mainly recommending movies that fall into the favored genres of users. It is worth noting

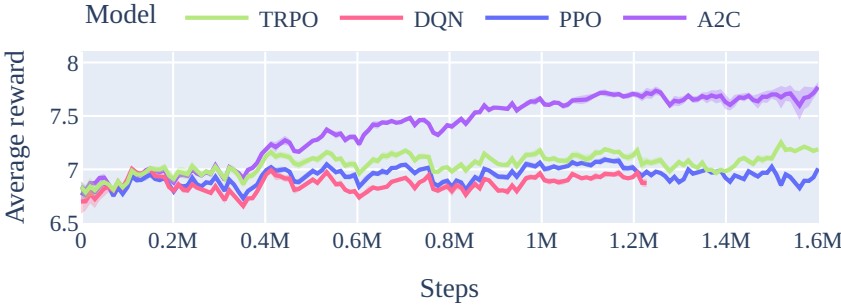

Figure 6: Training plot of various RL models. The y-axis displays the average reward from evaluation samples.

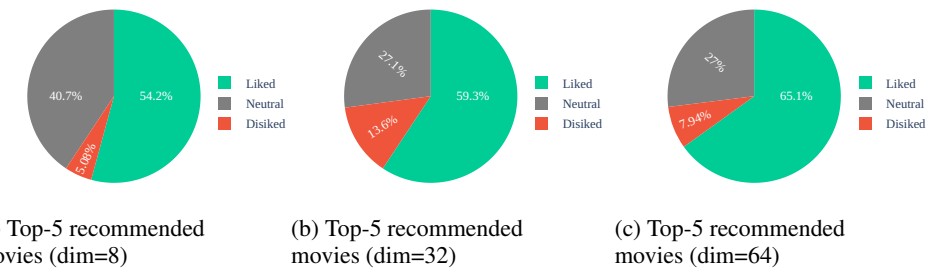

(a) Top-5 recommended movies (dim=8)

(b) Top-5 recommended movies (dim=32)

(c) Top-5 recommended movies (dim=64)

Figure 7: User genre preference statistic of top-5 movie recommendations generated by the RL model, with embedding dim 64 (8) and embedding dim (32).

that recommending a *neutral* movie can be a valid strategy, especially if it is a highly praised or outstanding movie.

### E.2 RECOMMENDED MOVIES EXAMPLES

For a set of random users we interact with the trained RL model, in this section with embedding dim 64, and show the first 5 recomended movies.

Table 15: At the top we show the personal interest of Max. At the bottom, we show the genres the first 5 recommended movies falls into for each recommended movie.

| Name | Max |
|---|---|
| **Liked genres** | thriller, documentary, fantasy, crime |
| **Disliked genres** | romance |
| **Recommended movie** | **Genres** |
| Wallace & Gromit: The Curse of the Were-Rabbit | adventure, animation, comedy, family |
| Mission: Impossible | adventure, action, thriller |
| Live Free or Die Hard | action, thriller |
| The World's Fastest Indian | drama, adventure, history |
| The Fugitive | action, thriller, drama |

Table 16: At the top we show the personal interest of Ava. At the bottom, we show the genres the first 5 recommended movies falls into for each recommended movie.

| **Name** | Ava |
|---|---|
| **Liked genres** | drama, science fiction, animation, adventure |
| **Disliked genres** | romance, fantasy, crime, comedy |
| **Recommended movie** | **Genres** |
| Batman: Assault on Arkham | thriller, animation, action, crime |
| Armour of God | adventure, action, comedy |
| The Magnificent Seven | adventure, action, western |
| Mission: Impossible | adventure, action, thriller |
| The World's Fastest Indian | drama, adventure, history |
| Gladiator | action, drama, adventure |

Table 17: At the top we show the personal interest of Maya. At the bottom, we show the genres o the first 5 recommended movies falls into for each recommended movie.

| **Name** | Maya |
|---|---|
| **Liked genres** | drama, science fiction, documentary, comedy |
| **Disliked genres** | horror, animation |
| **Recommended movie** | **Genres** |
| Jurassic Park | adventure, science fiction |
| The Way of the Dragon | action, crime |
| Master and Commander: The Far side of the World | adventure, drama, war |
| Mission: Impossible | adventure, action, thriller |
| Batman Begins | action, crime, drama |
| The Bourne Supremacy | action, drama, thriller |

### E.3 ENVIRONMENT PERFORMANCE

Table 18: List of Large Language Models tested on the environment. iterations/seconds are computed for all models using GPTQ and Exllama on a RTX3090, and A100-40GB for Llama-2-70B.

| Model name | Size | Contex length | iterations/s (in our env) |
|---|---|---|---|
| GPT-4 | 1760B | 8k / 32k | API ratelimit dependent |
| GPT-3.5 | 175B | 4k / 16k | |
| Llama-2-Chat | 70B | 4,096 | 1.6 |
| | 13B | | 5 |
| | 7B | | 6 |
| Vicuna-v1.3 | 33B | 2,048 | 3 |
| | 13B | | 5 |
| | 7B | | 6 |
| Vicuna-v1.5 | 13B | 4,096 | 5 |
| | 7B | | 6 |

## F    LIST OF HOBBIES AND PROFESSIONS

Table 19: Hobby of children

| Hobby-name |
| --- |
| Drawing and painting |
| Playing piano |
| Playing guitar |
| Playing violin |
| Playing flute |
| Playing drums |
| Dancing |
| Reading books |
| Writing stories |
| Board games |
| Card games |
| Gardening |
| Cooking |
| Backing |
| Building with Lego |
| Collecting stamps |
| Collecting coins |
| Collecting cards |
| Photography |
| Learning magic tricks |
| Soccer |
| Basketball |
| Swimming |
| Volleyball |
| Tennis |
| Acting |
| Singing |
| Puppetry |
| Birdwatching or nature exploration |
| Science experiments |
| Playing video games |
| Origami |
| Learning a new language |

Table 20: Jobs list

| Jobs |
| --- |
| Account Manager |
| Accountant |
| Actor |
| Actuary |
| Administrator |
| Advertising Executive |
| Aerospace Engineer |
| Aerospace Technician |
| Air Traffic Controller |
| Animal Trainer |
| Architect |
| Archivist |
| Art Director |
| Artist |
| Auctioneer |
| Auto Mechanic |
| Baggage Handler |
| Bailiff |
| Baker |
| Banker |
| Barber |
| Barber Shop Owner |
| Barista |
| Bartender |
| Benefits Administrator |
| Bicycle Mechanic |
| Biologist |
| Blacksmith |
| Boat Captain |
| Bodyguard |
| Bookkeeper |
| Botanical Illustrator |
| Botanist |
| Brewery Worker |
| Bricklayer |
| Broadcast Technician |
| Building Inspector |
| Bus Driver |
| Bus Mechanic |
| Butcher |
| CIO (Chief Information Officer) |
| Cabin Crew |
| Cake Decorator |
| Call Center Operator |
| Car Salesperson |
| Carpenter |
| Cartographer |
| Cashier |
| Casino Dealer |
| Caterer |

Table 21: Jobs list

Chaplain
Chauffeur
Chef
Chemical Engineer
Chemist
Chief Financial Officer (CFO)
Chimney Sweep
Chiropractor
Civil Engineer
Claims Adjuster
Cleaner
Clown
Coach
Coachbuilder
Commercial Pilot
Composer
Computer Programmer
Computer Systems Analyst
Concierge
Conservationist
Construction Manager
Construction Worker
Cost Estimator
Counselor
Courier
Court Reporter
Craftsperson
Cruise Ship Captain
Cryptographer
Curator
Customer Service Representative
Dairy Farmer
Dancer
Data Analyst
Data Entry Operator
Database Administrator
Demolition Worker
Dental Hygienist
Dentist
Designer
Desktop Publisher
Detective
Detective Inspector
Dialysis Technician
Diesel Mechanic
Dietician
Digital Marketer
Dispatch Operator
Doctor
Dog Trainer
Door-to-Door Salesperson
Dressmaker

Table 22: Jobs list

Drummer
Dry Cleaner
Economist
Economist
Electrician
Engineer
Event Planner
Farmer
Fashion Designer
Firefighter
Flight Attendant
Florist
Forensic Scientist
Gardener
Geologist
Graphic Designer
Hairdresser
Historian
Hotel Manager
Human Resources Manager
Illustrator
Industrial Designer
Insurance Agent
Interior Designer
Interpreter
Janitor
Journalist
Judge
Laboratory Technician
Lawyer
Librarian
Lifeguard
Linguist
Locksmith
Makeup Artist
Manager
Marketing Specialist
Massage Therapist
Mechanic
Medical Assistant
Meteorologist
Model
Musician
Nanny
Nurse
Nutritionist
Occupational Therapist
Optician
Painter
Paramedic
Pharmacist
Photographer
Physical Therapist
Physician Assistant
Pilot

Table 23: Jobs list

Plumber
Police Officer
Politician
Postal Worker
Producer
Professor
Psychologist
Public Relations Specialist
Real Estate Agent
Receptionist
Reporter
Research Scientist
Sales Representative
Scientist
Security Guard
Singer
Social Media Manager
Social Worker
Software Developer
Sound Engineer
Speech Therapist
Sports Coach
Statistician
Stockbroker
Surveyor
Tailor
Teacher
Technical Writer
Technician
Therapist
Tour Guide
Translator
Travel Agent
Truck Driver
UI/UX Designer
Veterinarian
Video Editor
Waiter/Waitress
Web Developer
Welder
Writer
Yoga Instructor
Zookeeper