# OpenReview forum: "SUBER: An RL Environment with Simulated Human Behavior for Recommender Systems"
_ICLR.cc/2024/Conference — Submitted to ICLR 2024_

### Official Review · Reviewer_7Rhx · 2023-10-28

**Soundness:** 1 poor
**Presentation:** 3 good
**Contribution:** 1 poor
**Rating:** 3
**Confidence:** 5

**Summary:**

The paper propose SUBER, a RL environment that applies LLM to simulate user behaviors. SUBER consists of several components including memory, preprocessing,  postprocessing, and LLM modules. The user history is sent to the RL module, and it returns an item. Both user history and the recommended item are processed as a prompt, then the LLM outputs the simulated user rating over the item. SUBER is built on two public datasets. The paper does experiments to validate its effectiveness. Finally, the paper shows that how A2C is trained in this environment.

**Strengths:**

1.Using LLM to simulate user behaviors is interesting.
2.The paper is easy to follow.
3.The paper does detailed ablation study.

**Weaknesses:**

1.The paper does not discuss the limitation of SUBER. In my opinion, I think the input feature is quite simple, and miss numerical and sequential informations.
2.The paper does not compare SUBER and other RL-based simulators, such as VirtualTaobao, RecoGym. Thus it is quite hard to evaluate the significance of SUBER in the RL4RS area.
3.The paper does not cite two recent papers about RL4RS simulators, "RL4RS: A Real-World Dataset for Reinforcement Learning based Recommender System" and KuaiSim: A Comprehensive Simulator for Recommender Systems.
4.As an RL environment, I would suggest that the author evaluate more RL algorithms besides A2C, such as SA2C(Supervised Advantage Actor-Critic for Recommender Systems), HAC(Exploration and Regularization of the Latent Action Space in Recommendation) and off-policy top-k(Top-K Off-Policy Correction for a REINFORCE Recommender System).

**Questions:**

See the above question.

---

> ### Author Response · Authors · 2023-11-19
>
> We thank the reviewer for their effort on reviewing our work. We address their concerns in the following.
>
> > The paper does not discuss the limitation of SUBER. In my opinion, I think the input feature is quite simple, and miss numerical and sequential informations.
>
> The feature space of our environment contains both numerical and sequential information, i.e., for each user we have a list of all movies previously watched together with their rating in sequential order.
>
> Since we wanted to demonstrate that a simple RL model can learn to recommend items in our environment, we use a simple state space that only records the last rating that the user gave to a movie. We chose this simpler scenario to show the learnability of our environment, and that the trained policy has some interesting features that we can interpret as human (cf. Appendix E.1, Figure 7). We have added more experiments on other RL algorithms in the revised manuscript (Section 4.3).
>
> It is also worth noting that the environment can be adapted easily and can support any type of feature.
>
> > The paper does not compare SUBER and other RL-based simulators, such as VirtualTaobao, Recogym. Thus it is quite hard to evaluate the significance of SUBER in the RL4RS area.
>
> The lack of comparisons with alternative environments is mainly because of the synthetic nature of our experimental setting. Thus, it is impractical to draw direct comparisons with other environments. Our environment is more closely related to Recogym [1] and Recsim [2] in this regard. We have provided a more extensive answer on this topic in the general response.
>
> > The paper does not cite two recent papers about RL4RS simulators, "RL4RS: A Real-World Dataset for Reinforcement Learning based Recommender System" and KuaiSim: A Comprehensive Simulator for Recommender Systems.
>
> Thank you for pointing out these two contemporary works, we have updated our related work section to include these in the revised manuscript.
>
> > As an RL environment, I would suggest that the author evaluate more RL algorithms besides A2C, such as SA2C(Supervised Advantage Actor-Critic for Recommendwiller Systems), HAC(Exploration and Regularization of the Latent Action Space in Recommendation) and off-policy top-k(Top-K Off-Policy Correction for a REINFORCE Recommender System).
>
> We have extended our analysis beyond the initial experiments with A2C, and added evaluations for additional RL algorithms (e.g., PPO, TRPO, DQN).
>
> Regarding the specific methods the reviewer mentioned (SA2C, HAC, top-k), we face practical limitations. These methods currently lack publicly available, gym-compatible interfaces, making direct comparison within our environment challenging. We acknowledge their importance and hope to include such methods in future work as they become more accessible.
>
> However, we would like to emphasize that our primary contribution lies in providing a versatile recommender system environment designed for experimenting with various RL and RecSys algorithms. The intent is to offer a platform for future explorations rather than conducting exhaustive evaluations of all existing algorithms.
>
> References:
>
> [1] Rohde, et al. (2018). Recogym: A reinforcement learning environment for the problem of product recommendation in online advertising.
>
> [2] Ie et al. (2019). Recsim: A configurable simulation platform for recommender systems.

---

### Official Review · Reviewer_TXpX · 2023-10-29

**Soundness:** 2 fair
**Presentation:** 2 fair
**Contribution:** 2 fair
**Rating:** 3
**Confidence:** 5

**Summary:**

The paper proposed SUBER, a framework designed to address common challenges in RL-based recommender systems, such as issues related to data availability and the design of reward functions. The paper conducted several ablation studies on movie and book recommendations to demonstrate the effectiveness of the method and examine the effect of each component in the framework.

**Strengths:**

-	The motivation of this paper is clear and the challenges it aims to address are significant to the RL-based recommender system.
-	The method is presented clearly. Each component is well explained, and the flow of the entire framework is well presented in the figure.
-	The paper conducted a series of ablation studies to scrutinize the effect of different components within the framework.

**Weaknesses:**

-	The proposed  framework is to tackle key challenges in RL4Rec, such as data accessibility, the uncertainty of the user model, and the assessment of models. However, the originality of this research is ambiguous to me. It appears to predominantly integrate components of Reinforcement Learning (RL) with Large Language Models (LLMs). Furthermore, due to the absence of a thorough comparison with existing RL simulators and state-of-the-art techniques including RL4Rec and LLM-integrated RecSys, it's challenging to position the precise significance of the contributions claimed in this study.
-	The paper has only made comparisons between different settings of SUBER on the metrics proposed in this paper. The comparisons with other methods on some commonly used metrics such as MAP/R^2/Personalization are missing. These comparisons would be essential to understand the benefits of using this method.
-	The prompts used in the pre-processing module and the user description generation step require hand-crafted templates, which may limit the generalizability of the method in other scenarios.
-	This method may require more computational resources than other methods due to the usage of LLM. More analysis and evaluations should be done.
-	The authors failed to monitor significant existing literature on RL4Rec, including various methods and simulators, as well as RecSys integrated with LLMs. This oversight renders the paper's scope and credibility questionable.

**Questions:**

-	The paper claims that it addresses the challenge of model evaluation, is it referring to the evaluation metrics mentioned in Section 4.2 and Table 1-2? How do these metrics outperform the existing evaluation methods?
-	As addressed in the weakness section, I think the comparisons between this method and other methods on metrics such as MAP/R^2/Personalization are essential to verify the effectiveness of the method. Could the authors provide these results?
-	I’m not sure about the purpose of generating the user descriptions. The paper mentioned that the Age / Job / Hobbies of the users are randomly sampled from external distributions, how does this random information affect the outcome? And what’s the motivation for doing so? An ablation study to compare the results with/without this information would be helpful.
-	It’s known that different prompt methods could affect the response of LLMs. How does the prompt template affect the outcomes in this framework? I suggest the authors try several different prompt templates in the pre-processing module and the user description generation step, then report the range of the results.
-	In Fig 4a, I observed a significant performance drop somewhere between 1.6M - 1.7M steps, why does this happen? It seems not due to the randomness because this pattern is consistent across all embedding dimensions. Or more generally, I found the pattern of these three lines seems to be extremely similar, I’m surprised by this because I suppose these are three independent experiments with different dimension settings. Is there any particular reason for the similarity between these three lines?

---

> ### Author Response · Authors · 2023-11-19
>
> We thank the reviewer for their effort in reviewing our work. We address their concerns in the following.
>
> > The proposed framework is to tackle key challenges in RL4Rec, such as data accessibility, the uncertainty of the user model, and the assessment of models. However, the originality of this research is ambiguous to me. It appears to predominantly integrate components of Reinforcement Learning (RL) with Large Language Models (LLMs). Furthermore, due to the absence of a thorough comparison with existing RL simulators and state-of-the-art techniques including RL4Rec and LLM-integrated RecSys, it's challenging to position the precise significance of the contributions claimed in this study.
>
> All previous related work has either incorporated LLMs as the RecSys model itself, or used other strategies (e.g., GAN, transformers, statistical models) to simulate realistic user behavior. LLMs have been shown to excel in simulating user behavior through natural language [1,2].
>
> The main novelty of our work is that we tackle the challenges in RL4Rec by introducing LLMs into the environment on which the RL RecSys algorithm is trained. This approach allows us to create a synthetic environment which simulates user behavior in a more natural and realistic manner. Furthermore, our environment provides practitioners the ability to train and evaluate their RL RecSys agents online.
>
> We have added a table and updated the related work in the revised manuscript to provide a more in-depth comparison with existing research.
>
> > The paper has only made comparisons between different settings of SUBER on the metrics proposed in this paper. The comparisons with other methods on some commonly used metrics such as MAP/R^2/Personalization are missing. These comparisons would be essential to understand the benefits of using this method.
>
> We utilize LLMs within a simulation environment. Our main contribution is the simulation environment rather than an LLM-based agent. Thus, comparing our environment with others based on metrics like MAP, MSE, and personalization is not feasible. We updated the related work and introduction in the revised manuscript to make this distinction more clear. Furthermore, we have included benchmarks for more RL methods that were trained on our environment, see Section 4.3.
>
> It is generally not feasible to compare our framework with others due to the synthetic nature of our environment. We provide a more detailed answer in the general response.
>
> > The prompts used in the pre-processing module and the user description generation step require hand-crafted templates, which may limit the generalizability of the method in other scenarios.
>
> We have constructed a general prompt that works for, but is not limited to, both movie and book recommendation, where we change only specific wording such as “movie” with “books”.
> In terms of generalizability, this is not an issue. The prompting component can be freely customized to use different types of items in different settings. The environment is designed with modularity in mind, and therefore, adapting different components for different scopes is feasible. In fact, the prompt used in all of our experiments is not fixed and can be changed at will.
>
> > This method may require more computational resources than other methods due to the usage of LLM. More analysis and evaluations should be done.
>
> We performed benchmarks to understand the throughput (interactions/second) that our environment can achieve (see Table 18 in Appendix E.3 for a single GPU). It is also worth noting that the environment can be accelerated using multiple GPUs in parallel. In order to achieve the interactions/second mentioned in Table 18, we also engineered a specific prompt structure, which takes advantage of caching and significantly speeds up (4x) the performance of the environment (cf. paragraph 2 section 4.1).
>
>
> > The authors failed to monitor significant existing literature on RL4Rec, including various methods and simulators, as well as RecSys integrated with LLMs. This oversight renders the paper's scope and credibility questionable.
>
> We have added Recogym [3], Recsim [4], RL4RS [5], and KuaiSim [6] in the related work and discussed how our approach compares to these works, the details of which can be found in our general response.

---

> ### Author Response · Authors · 2023-11-19
>
> > The paper claims that it addresses the challenge of model evaluation, is it referring to the evaluation metrics mentioned in Section 4.2 and Table 1-2?
>
> Table 4 in Section 4.2 and Table 13 in Appendix D.2 (before Table 1-2) are not tools for evaluating RL models; rather, they serve to qualitatively assess our environment's capability to mimic human concepts (e.g., genre preference, item collection), via natural language analysis.
>
> The challenge of model evaluation in our paper specifically pertains to the inherent complexities of evaluating an RL model in RL4Rec, where one trained model is to be tested on live, online data rather than relying on pre-recorded, logged data.
>
> We address this challenge by developing our synthetic environment on which RL models can be tested online.
>
> > How do these metrics outperform the existing evaluation methods?
>
> These metrics are unique to our environment and are employed internally for the evaluation of the environment and its different modular components, rather than the trained RL RecSys model. Hence, they are not meant for direct comparison with the existing evaluation metrics used by other related work.
>
> > As addressed in the weakness section, I think the comparisons between this method and other methods on metrics such as MAP/R^2/Personalization are essential to verify the effectiveness of the method. Could the authors provide these results?
>
> In the updated manuscript, we tested various RL algorithms trained in our proposed environment on common recommendation metrics, such as MAP@10, MMR@10, and personalization.
>
> > I’m not sure about the purpose of generating the user descriptions. The paper mentioned that the Age / Job / Hobbies of the users are randomly sampled from external distributions, how does this random information affect the outcome? And what’s the motivation for doing so? An ablation study to compare the results with/without this information would be helpful.
>
> Generating the user description is important because LLMs work with textual data. The LLM is at the core of our environment and is used to simulate users. The RL model interacts with the environment by interacting with these simulated users that are part of the environment.
>
> The user description is generated by the LLM. Note that the dataset generation is done a priori and then stored in memory, it is not done during the RL training. Some basic additional information is provided to the LLM to generate the user description, such as age/profession/hobbies. The generated user description will strongly depend on this information, as it is included in the prompt when generating the user. The main goal of this process is to generate users that are different from each other; if the users are generated randomly by the LLM, they will suffer from some inherent biases of the language model. We found that users generated by the LLM without providing additional information were very similar to each other, and therefore, not diverse enough for an environment. In contrast, LLMs generating users using this additional information do not suffer from this, and the generated users represented a diverse group of humans (cf. Appendix A.1 Figure 4).

---

> ### Author Response · Authors · 2023-11-19
>
> > It’s known that different prompt methods could affect the response of LLMs. How does the prompt template affect the outcomes in this framework? I suggest the authors try several different prompt templates in the pre-processing module and the user description generation step, then report the range of the results.
>
> We are aware that prompting is very important and different templates can influence the outcome of the LLM. Therefore, we compared various different prompt templates (Table 4 in Section 4.2 and Table 13 in Appendix D.2). We experimented with few shots prompting, shifting the range of the rating scale as well as custom system prompts. It is worth mentioning that during the development of our environment we tested a multitude of different prompt strategies, however, we only present the most relevant results in our work.
> Additionally, we also worked on the prompt structure to enhance the performance of our environment (Section 4.1). The prompts are constructed such that we can leverage the key-value cache of LLMs to speed up (4x) the interactions/second of our environment and reduce the time needed to train an RL model on it.
>
> > In Fig 4a, I observed a significant performance drop somewhere between 1.6M - 1.7M steps, why does this happen? It seems not due to the randomness because this pattern is consistent across all embedding dimensions. Or more generally, I found the pattern of these three lines seems to be extremely similar, I’m surprised by this because I suppose these are three independent experiments with different dimension settings. Is there any particular reason for the similarity between these three lines?
>
> This phenomenon was due to the same seed used across the experiments for reproducibility and consistency. We have added more experiments with multiple seeds and computed the confidence intervals for all of our testing. As shown in Appendix E.1 Figure 6 in the revised manuscript, we no longer have such a significant performance drop. We have updated the revised manuscript with the new results.
>
> References:
>
> [1] Park et al. (2023). Generative agents: Interactive simulacra of human behavior.
>
> [2] Argyle et al. (2023). Out of one, many: Using language models to simulate human samples
>
> [3] Rohde, et al. (2018). Recogym: A reinforcement learning environment for the problem of product recommendation in online advertising.
>
> [4] Ie et al. (2019). Recsim: A configurable simulation platform for recommender systems.
>
> [5] Wang et al. (2021). Rl4rs: A real-world benchmark for reinforcement learning based recommender system
>
> [6] Zhao et al. (2023). KuaiSim: A comprehensive simulator for recommender systems.

---

### Official Review · Reviewer_K35w · 2023-11-04

**Soundness:** 3 good
**Presentation:** 3 good
**Contribution:** 3 good
**Rating:** 5
**Confidence:** 4

**Summary:**

This paper proposes a framework for training and evaluating RL-based recommender systems, which uses large language models (LLMs) to simulate human behavior and rate recommended items.

**Strengths:**

+ The paper proposes a user simulation framework based on large language models, which can alleviate the problems of data scarcity and model evaluation for reinforcement learning based recommender system.
+ The paper designs a flexible and extensible environment based on reinforcement learning principles, which can interact with different LLMs and recommendation strategies.
+ The paper provides a modular framework and open-source code, which is a valuable tool for the recommender system domain. It helps researchers and developers to train and evaluate reinforcement learning based recommender systems without real user interactions.

**Weaknesses:**

- The manuscript could benefit from more robust experimental support. The absence of comparisons with other simulation algorithms may impact the persuasiveness of the paper.

- The paper appears to lack some key experiments that validate the effectiveness and advantages of RL training in the proposed environment. While the paper presents ablation studies on different components of the environment, it does not compare the RL-based recommender system with other baselines or state-of-the-art methods on real user data or benchmark datasets. It would be advantageous if the authors could include such experimental validations in future research.

- The use of LLMs to generate synthetic users is an innovative approach that leverages the powerful capabilities of LLMs to simulate human behavior and preferences. However, the paper does not evaluate the quality and diversity of the user generation, nor does it compare it with real user data. This could potentially lead to biases and inaccuracies in the simulation. It would be beneficial if the authors could delve deeper into this aspect in future research.

- The related work section of the paper seems too general and lacks precision. It does not adequately highlight the differences and connections between their work and existing research. It would be advantageous if the authors could elaborate more on the relationship and uniqueness of their work in relation to existing research, to enhance the depth and breadth of the paper.

**Questions:**

1.	Can the author provide a distribution chart for scores predicted by LLMs and actual scores? The numbers reported in the table do not provide an intuitive image. On the other hand, it is more important to focus not on the overall score distribution, but on the differences in scoring for each item (it is possible for the overall score distribution to be the same, but with significant differences in individual item scores).
2.	What is the difference between the last two rows in Table 1? Is it a type error?
3.	Comparing rows 3-8 in Table 1 with the 9th row, it can be observed that the model is very good at distinguishing between High and Low Ratings on a scale of 0-9. However, when the score scale changes to 1-10, there is a significant decrease in performance. Does this indicate that the model can only identify very poor items? (0 score)
4.	I suggest the author showcases the performance of several traditional models and conventional RL methods, under your reward metric (Figure .4a)

---

> ### Author Response · Authors · 2023-11-19
>
> We thank the reviewer for their effort on reviewing our work. We address their concerns in the following.
>
> > The manuscript could benefit from more robust experimental support. The absence of comparisons with other simulation algorithms may impact the persuasiveness of the paper.
>
> The lack of comparisons with other environments is mainly due to the synthetic nature of our environment. As a result, comparing our environment with others is not feasible. From this perspective, we are more similar to Recogym [1] and Recsim [2]. We have provided more details in the general response.
>
> > The paper appears to lack some key experiments that validate the effectiveness and advantages of RL training in the proposed environment. While the paper presents ablation studies on different components of the environment, it does not compare the RL-based recommender system with other baselines or state-of-the-art methods on real user data or benchmark datasets. It would be advantageous if the authors could include such experimental validations in future research.
>
> We have added additional experiments with different RL models in our environment, in particular A2C, PPO, DQN, and TRPO. In Section 4.3, we demonstrated that A2C can effectively learn in our environment by achieving a high average reward, MAP@10, and MRR@10 score. Additionally, we also demonstrate random qualitative examples. Appendix E further illustrates how A2C is able to recommend movies according to the user's genre preferences.
>
> > The use of LLMs to generate synthetic users is an innovative approach that leverages the powerful capabilities of LLMs to simulate human behavior and preferences. However, the paper does not evaluate the quality and diversity of the user generation, nor does it compare it with real user data. This could potentially lead to biases and inaccuracies in the simulation. It would be beneficial if the authors could delve deeper into this aspect in future research.
>
> We agree with the reviewer that the use of LLMs could potentially lead to bias. To mitigate this, we reduce the bias by explicitly mentioning in the system prompt that the model should not be biased and should answer the prompts realistically (Appendix A.3.1). Furthermore, we sampled characteristics of users like professions, hobbies, age, and interests. This process ensures the LLM creates unbiased synthetic users (e.g., not biased on age or interests) (Appendix A.1). Hobbies and professions are randomly sampled from a list, while age and interests are sampled from real human distributions (Appendix A.1). These choices are included in the prompt, so when the LLM generates the user description, it is conditioned on the previously sampled characteristics. This allows for a more diverse distribution of users, rather than being biased towards a particular group of people. We show the effectiveness of our approach in Figure 4 (Appendix A.1).
>
> > The related work section of the paper seems too general and lacks precision. It does not adequately highlight the differences and connections between their work and existing research. It would be advantageous if the authors could elaborate more on the relationship and uniqueness of their work in relation to existing research, to enhance the depth and breadth of the paper.
>
> We have added a table and updated the related work section to provide a more in-depth comparison with existing research.

---

> ### Author Response · Authors · 2023-11-19
>
> > Can the author provide a distribution chart for scores predicted by LLMs and actual scores? The numbers reported in the table do not provide an intuitive image. On the other hand, it is more important to focus not on the overall score distribution, but on the differences in scoring for each item (it is possible for the overall score distribution to be the same, but with significant differences in individual item scores).
>
> We have added a graph of the score probability distribution in Figure 5 (Appendix D.1) of the revised manuscript.
>
> > What is the difference between the last two rows in Table 1? Is it a type error?
>
> Not an error, these are two different strategies of prompting, in one case “1-10” refers to the digits between “1” and “10” in the prompt, while “one-ten” refers to ratings in the prompt which were written in words (“one”, “two”, “three”, ..., “ten”) (cf. 4th paragraph section 4.1). We have updated the table captions to better reflect this in the revised manuscript.
>
> > Comparing rows 3-8 in Table 1 with the 9th row, it can be observed that the model is very good at distinguishing between High and Low Ratings on a scale of 0-9. However, when the score scale changes to 1-10, there is a significant decrease in performance. Does this indicate that the model can only identify very poor items? (0 score)
>
> We found this to be an interesting problem. We believe this is related to the ambiguity of the tokenization, however, we were unable to find any research discussing this issue. In general, the 0-9 scale performs much better than the 1-10 scale in almost all of our experiments, mainly because the token “10”, which represents the best score, can be tokenized in 2 ways, either the token “10” directly or the token “1” and then the token “0” (cf. 4th paragraph section 4.1).
> In these experiments, the 0-9 scale outperforms the 1-10 scale significantly. This is because, in the 0-9 scale, when the language model assigns a high rating, it directly corresponds to the token “9”. In contrast, in the 1-10 scale, the model occasionally generates the rating “10” by producing the digits “1” and “0” separately, whereas, in some cases, the model generates the EOS-token after computing the token “1”, leading the model to obtain a high loss since it predicted score (1) compared to the true score (10).
>
> > I suggest the author showcases the performance of several traditional models and conventional RL methods, under your reward metric (Figure .4a)
>
> In the revised manuscript, we added more experiments on training RL methods like PPO, A2C, DQN and TRPO. The experiments can be found in Section 4.3.
>
> References:
>
> [1] Rohde, et al. (2018). Recogym: A reinforcement learning environment for the problem of product recommendation in online advertising.
>
> [2] Ie et al. (2019). Recsim: A configurable simulation platform for recommender systems.

---

### Official Review · Reviewer_g6on · 2023-11-11

**Soundness:** 2 fair
**Presentation:** 3 good
**Contribution:** 2 fair
**Rating:** 6
**Confidence:** 5

**Summary:**

The authors present a promising solution to the challenge of training recommender systems when real user interactions are not available. They propose SUBER, a novel Reinforcement Learning (RL) simulated environment tailored for recommender system training, which leverages recent advancements in Large Language Models (LLMs) to simulate human behavior within the training setting. A series of ablation studies and experiments demonstrate the effectiveness of their approach. This research represents a significant step towards creating more realistic and practical training environments for recommender systems, even in the absence of direct user interactions.

**Strengths:**

1.	The concept of employing Large Language Models (LLMs) to mimic synthetic users is intriguing. SUBER offers a multifaceted approach, generating synthetic data while harnessing the potential of LLMs to accurately emulate the behavior of users with undisclosed patterns.
2.	The authors meticulously conduct comprehensive ablation studies to dissect the various components of LLMs, demonstrating the scalability and versatility of SUBER in the process.
3.	The article is impeccably articulated, presenting its ideas with clarity and maintaining a logical flow throughout.

**Weaknesses:**

1.	How do the authors assess the accuracy of their simulated environment? The paper primarily showcases the training curve of the RL model within this environment but lacks a comparative analysis against other environment simulation methods. The absence of online experiments further challenges the validity of the simulated environment.

2.	The authors should expound on their rationale for using LLMs to simulate users, clarify why this approach is effective, and outline the advantages it offers over alternative environment simulation methods.

3.	The authors claim that this dynamic environment can serve as a model evaluation tool for recommender systems, but it lacks empirical evidence to support this claim. A clear methodology for measuring the accuracy of the proposed evaluation method is needed.

4.	The absence of t-tests or error bars in the results section raises concerns about the reliability and reproducibility of the experimental findings.

5.	The paper mentions the limitation of context length for providing a list of all possible items to an LLM. Further details are required regarding how the author addressed this particular issue.

**Questions:**

1.	I suggest the authors evaluate the simulated environment from more aspects. To establish the accuracy of the simulated environment, consider conducting comparative experiments. Compare the performance of your proposed environment with existing methods for simulating user interactions. Additionally, performing online experiments where applicable, could help validate the authenticity of your simulated environment.

2.	I suggest the authors provide a more in-depth explanation of why LLMs are chosen to simulate users. Elaborate on the effectiveness of this approach by highlighting its advantages over alternative simulation methods. This could include discussing how LLMs can capture complex user behavior or adapt to changing patterns more effectively.

3.	To substantiate the claim that your dynamic environment serves as a model evaluation tool, conduct experiments that demonstrate its utility in evaluating recommender systems. Present a clear experimental setup and results that support this assertion.

4.	Enhance the reliability and reproducibility of your experimental results by including t-tests or error bars.

5.	Explain in detail how you addressed the limitation of limited context length. What techniques or strategies did you employ to mitigate this constraint when using LLMs in your environment?

---

> ### Author Response · Authors · 2023-11-19
>
> We thank the reviewer for their effort on reviewing our work. We address their concerns in the following.
>
> > How do the authors assess the accuracy of their simulated environment? The paper primarily showcases the training curve of the RL model within this environment but lacks a comparative analysis against other environment simulation methods. The absence of online experiments further challenges the validity of the simulated environment.
>
> The absence of any comparisons with other environments is primarily due to the synthetic nature of our environment. Unfortunately, comparing our environment with others is unfeasible. In this regard, our environment bears more similarity to Recogym [1] and Recsim [2]. We have provided more details on this matter in the general response.
>
>
> > The authors should expound on their rationale for using LLMs to simulate users, clarify why this approach is effective, and outline the advantages it offers over alternative environment simulation methods.
>
> The main motivation for using LLMs is their ability to simulate human behavior well, as demonstrated by previous work [3, 4]. Therefore, such an environment is able to capture human dynamics and interests. We have updated the introduction of the revised manuscript to emphasize this aspect. Furthermore, LLMs undergo training on extensive datasets that include information related to various subjects. This enhances their capability to grasp context, and therefore, respond in a realistic manner.
>
>
> > The authors claim that this dynamic environment can serve as a model evaluation tool for recommender systems, but it lacks empirical evidence to support this claim. A clear methodology for measuring the accuracy of the proposed evaluation method is needed.
>
> We conduct additional experiments using different RL methods within our proposed environment (Section 4.3). This environment is based on user ratings - in other words, when a movie is recommended to a user, the RL model receives a reward corresponding to the user's rating for that movie. Consequently, the average reward accumulated by the RL model serves as a key metric for evaluating its performance in the environment.
>
> As shown in the experimental results described in Section 4.3, we can measure the effectiveness of the RL model by monitoring its average reward. A higher average reward indicates that the model has successfully selected movies that match interests of users. In addition to average reward, we have added additional metrics such as MAP@10, MMR@10, and personalization to provide a comprehensive evaluation of different RL models.
>
> > The absence of t-tests or error bars in the results section raises concerns about the reliability and reproducibility of the experimental findings.
>
> We have added error bars to the figures and tables in the revised manuscript.
>
> > The paper mentions the limitation of context length for providing a list of all possible items to an LLM. Further details are required regarding how the author addressed this particular issue.
>
> To address the limitation of a fixed context length, we have implemented the Item Retrieval component. For a given user, when constructing a prompt, the Item Retrieval component selects only a subset of items previously seen by the user. We mention this aspect and the role of the Item Retrieval component in Section 3.2.

---

> ### Author Response · Authors · 2023-11-19
>
> > I suggest the authors evaluate the simulated environment from more aspects. To establish the accuracy of the simulated environment, consider conducting comparative experiments. Compare the performance of your proposed environment with existing methods for simulating user interactions. Additionally, performing online experiments where applicable, could help validate the authenticity of your simulated environment.
>
> To the best of our knowledge, there is no practical and accurate method for comparing synthetic environments. We address this topic in more detail in the general response. If the reviewer holds a different perspective on this matter, we welcome the opportunity to engage in further discussion.
>
> > I suggest the authors provide a more in-depth explanation of why LLMs are chosen to simulate users. Elaborate on the effectiveness of this approach by highlighting its advantages over alternative simulation methods. This could include discussing how LLMs can capture complex user behaviour or adapt to changing patterns more effectively.
>
> We have added more qualitative explanations why LLMs are good for simulating users in the revised manuscript. We base ourselves on the work by Park et al. [3] and Argyle et al. [4] which show that LLMs are good simulators of human behavior.
>
> > To substantiate the claim that your dynamic environment serves as a model evaluation tool, conduct experiments that demonstrate its utility in evaluating recommender systems. Present a clear experimental setup and results that support this assertion.
>
> We added more experiments in Section 4.3 to compare different RL methods trained on SUBER.
>
> As shown in the experimental results described in Section 4.3, we can measure the effectiveness of the RL model by monitoring its average reward. A higher average reward indicates that the model has successfully selected movies that match interests of users. In addition to average reward, we have added additional RecSys specific metrics such as MAP@10, MMR@10, and personalization to provide a comprehensive evaluation of different RL models.
>
> > Enhance the reliability and reproducibility of your experimental results by including t-tests or error bars.
>
> We have conducted more experiments and added error bars and confidence intervals in the experiment section of the revised manuscript.
>
> > Explain in detail how you addressed the limitation of limited context length. What techniques or strategies did you employ to mitigate this constraint when using LLMs in your environment?
>
> We implemented three types of item retrieval: Recency Retrieval, in other words we retrieve the most recently seen items, Feature Similarity Retrieval based on the movie features, and T5 Similarity Retrieval based on the movie description. See Section 3.2 and Section 4.1 for more details. Furthermore, we compare these retrieval approaches against each other in Section 4.2.
>
> References:
>
> [1] Rohde, et al. (2018). Recogym: A reinforcement learning environment for the problem of product recommendation in online advertising.
>
> [2] Ie et al. (2019). Recsim: A configurable simulation platform for recommender systems.
>
> [3] Park et al. (2023). Generative agents: Interactive simulacra of human behavior.
>
> [4] Argyle et al. (2023). Out of one, many: Using language models to simulate human samples

---

### Author Response · Authors · 2023-11-19

We would like to express our sincere gratitude to the reviewers for their thoughtful and constructive feedback. We appreciate the time and effort they have dedicated to reviewing our work.

We are delighted to see that reviewers agree with our assessment that our approach “can alleviate the problems of data scarcity and model evaluation“ and is “significant to the RL-based recommender system." We are also glad to hear that “LLM to simulate user behaviors is interesting“ and “employing LLMs to mimic synthetic users is intriguing.”

In response to the feedback provided, we have carefully considered each suggestion and made corresponding revisions to address the concerns raised. Below, we outline the general improvements made in light of the reviewers' input.


Changes to the revised manuscript:
- Added additional experiments comparing RL algorithms trained on our environment on common RecSys metrics.
- Updated tables and figures with more information (e.g., confidence intervals, error bars)
- Added a table comparing our environment to existing related work.
- Clarified and improved various sections of the manuscript.

Reviewers noted that our work lacks a comparison with other environments. We understand that this aspect is crucial for a comprehensive evaluation; however, comparing synthetic environments presents challenges.

Our environment SUBER, while leveraging real data for items (such as movie information from MovieLens), uses synthetically generated users, making it difficult to quantitatively compare with other environments due to the absence of shared users between them.


Since our environment leverages the capabilities of LLMs, and is not trained on any sort of logging data, we cannot compare our environment to an environment trained on real data (e.g., KuaiSim [1], RL4RS [2]).

Similar to the methodologies used in Recogym [3] and Recsim [4], we focus on qualitative evaluations of SUBER through case studies and sanity checks.
Specifically, we evaluate the quality of SUBER by testing the understanding of genres and collection of items, as well as the high-low consistency test (Section 4.2).

Further, we demonstrate learnability by successfully training RL models in SUBER. We show that the trained RL models are able to understand the genre preferences of users even if they had no access to the user details (Appendix E.1).

References:

[1] Zhao et al. (2023). KuaiSim: A comprehensive simulator for recommender systems.

[2] Wang et al. (2021). Rl4rs: A real-world benchmark for reinforcement learning based recommender system

[3] Rohde, et al. (2018). Recogym: A reinforcement learning environment for the problem of product recommendation in online advertising.

[4] Ie et al. (2019). Recsim: A configurable simulation platform for recommender systems.

---

### Meta-Review · Area_Chair_vzYe · 2023-12-14

**Metareview:**

The paper proposed SUBER, a simulated environment for RL based recommendations by leveraging Large Language Models (LLMs) to simulate human behavior.  The proposal offers a promising solution to training recommender systems when real user interactions are missing.

Strength: leveraging LLMs to simulate users is intriguing. The paper is clearly written. The authors conducted a series of ablation studies to examine the effect of different components in the framework.
Weakness: missing comparison to other simulated environments. missing quality evaluation on the LLM based user simulation.

**Justification For Why Not Higher Score:**

While all the reviewers find the approach to leverage LLMs to simulate user behaviors intriguing and appreciate the ablation studies, the lack of quality evaluation on the synthetic generation and missing comparison to other simulated environments are critical concerns that need to be addressed before the paper can be accepted at this venue.

**Justification For Why Not Lower Score:**

N/A

---

### Decision · Program_Chairs · 2024-01-16

Reject